# Tenectin recruits integrin to stabilize bouton architecture and regulate vesicle release at the *Drosophila* neuromuscular junction

Qi Wang[1], Tae Hee Han[1], Peter Nguyen[1], Michal Jarnik[2], Mihaela Serpe[1]*

[1]Section on Cellular Communication, Eunice Kennedy Shriver, National Institute of Child Health and Human Development, National Institutes of Health, Bethesda, United States; [2]Section on Intracellular Protein Trafficking, Eunice Kennedy Shriver, National Institute of Child Health and Human Development, National Institutes of Health, Bethesda, United States

**Abstract** Assembly, maintenance and function of synaptic junctions depend on extracellular matrix (ECM) proteins and their receptors. Here we report that Tenectin (Tnc), a Mucin-type protein with RGD motifs, is an ECM component required for the structural and functional integrity of synaptic specializations at the neuromuscular junction (NMJ) in *Drosophila*. Using genetics, biochemistry, electrophysiology, histology and electron microscopy, we show that Tnc is secreted from motor neurons and striated muscles and accumulates in the synaptic cleft. Tnc selectively recruits $\alpha$PS2/$\beta$PS integrin at synaptic terminals, but only the *cis* Tnc/integrin complexes appear to be biologically active. These complexes have distinct pre- and postsynaptic functions, mediated at least in part through the local engagement of the spectrin-based membrane skeleton: the presynaptic complexes control neurotransmitter release, while postsynaptic complexes ensure the size and architectural integrity of synaptic boutons. Our study reveals an unprecedented role for integrin in the synaptic recruitment of spectrin-based membrane skeleton.
DOI: https://doi.org/10.7554/eLife.35518.001

*For correspondence:
mihaela.serpe@nih.gov

**Competing interests:** The authors declare that no competing interests exist.

## Introduction

The extracellular matrix (ECM) and its receptors impact every aspect of neuronal development, from axon guidance and migration to formation of dendritic spines and neuromuscular junction synaptic junctions and function. The heavily glycosylated ECM proteins provide anchorage and structural support for cells, regulate the availability of extracellular signals, and mediate intercellular communications (*Reichardt and Prokop, 2011*). Transmembrane ECM receptors include integrins, syndecans and the dystrophin-associated glycoprotein complex (*Bökel and Brown, 2002*; *Häcker et al., 2005*; *Waite et al., 2009*). Integrins in particular are differentially expressed and have an extensive repertoire, controlling multiple processes during neural development. In adults, integrins regulate synaptic stability and plasticity (*Morini and Becchetti, 2010*; *McGeachie et al., 2011*). However, integrin roles in synapse development have been obscured by their essential functions throughout development. How integrins are selectively recruited at synaptic junctions and how they engage in specific functions during synapse development and homeostasis remain unclear.

One way to confer specificity to ECM/integrin activities is to deploy specialized ECM ligands for the synaptic recruitment and stabilization of selective heterodimeric integrin complexes (*Reichardt and Tomaselli, 1991*). For example, at the vertebrate NMJ, three laminins containing the $\beta2$ subunit (laminin 221, 421 and 521, that are heterotrimers of $\alpha2/4/5$, $\beta2$ and $\gamma1$ subunits) are

**eLife digest** Nerve cells or neurons can communicate with each other by releasing chemical messengers into the gap between them, the synapse. Both neurons and synapses are surrounded by a network of proteins called the extracellular matrix, which anchors, protects and supports the synapse. The matrix also helps to regulate the dynamic communication across the synapses and consequently neurons.

Little is known about the proteins of the extracellular matrix, in particular about the ones involved in structural support. This is especially important for the so-called neuromuscular junctions, where neurons stimulate muscle contraction and trigger vigorous movement. Receptor proteins on cell surfaces, such as integrins, can bind to the extracellular matrix proteins to anchor the cells and are important for all cell junctions, including synaptic junctions. But because of their many essential roles during development, it was unclear how integrins modulate the activity of the synapse.

To investigate this further, Wang et al. studied the neuromuscular junctions of fruit flies. The experiments revealed that both muscle and neurons secrete a large protein called Tenectin, which accumulates into the small space between the neuron and the muscle, the synaptic cleft. This protein can bind to integrin and is necessary to support the neuromuscular junction structurally and functionally.

Wang et al. discovered that Tenectin works by gathering integrins on the surface of the neuron and the muscle. In the neuron, Tenectin forms complexes with integrin to regulate the release of neurotransmitters. In the muscle, the complexes provide support to the synaptic structures. However, when Tenectin was experimentally removed, it only disrupted the integrins at the neuromuscular junction, without affecting integrins in other regions of the cells, such as the site where the muscle uses integrins to attach to the tendon. Moreover, without Tenectin an important intracellular scaffolding meshwork that lines up and reinforces cell membranes was no longer organized properly at the synapse.

A next step will be to identify the missing components between Tenectin/integrin complexes on the surface of neurons and the neurotransmitter release machinery inside the cells. The extracellular matrix and its receptors play fundamental roles in the development and function of the nervous system. A better knowledge of the underlying mechanisms will help us to better understand the complex interplay between the synapse and the extracellular matrix.

DOI: https://doi.org/10.7554/eLife.35518.002

deposited into the synaptic cleft and basal lamina by skeletal muscle fibers and promote synaptic differentiation. However, only laminin 421 interacts directly with presynaptic integrins containing the α3 subunit and anchors a complex containing the presynaptic $Ca_v\alpha$ and cytoskeletal and active zone-associated proteins (*Carlson et al., 2010*). Studies with peptides containing the RGD sequence, recognized by many integrin subtypes, have implicated integrin in the morphological changes and reassembly after induction of long-term potentiation (LTP) (reviewed in [*McGeachie et al., 2011*]). Several integrin subunits (α3, α5, α8, β1 and β2) with distinct roles in the consolidation of LTP have been identified, but the relevant ligands remain unknown.

*Drosophila* neuromuscular junction (NMJ) is a powerful genetic system to examine the synaptic functions of ECM components and their receptors. In flies, a basal membrane surrounds the synaptic terminals only in late embryos; during development, the boutons 'sink' into the striated muscle, away from the basal membrane (*Prokop et al., 1998*). The synaptic cleft relies on ECM to withstand the mechanical tensions produced by the muscle contractions. The ECM proteins, including laminins, tenascins/teneurins (Ten-a and -m) and Mind-the-gap (Mtg), interact with complexes of five integrin subunits (αPS1, αPS2, αPS3, βPS, and βν) (*Broadie et al., 2011*). The αPS1, αPS2 and βPS subunits localize to pre- and post-synaptic compartments and have been implicated in NMJ growth (*Beumer et al., 1999*; *Beumer et al., 2002*). The αPS3 and βν are primarily presynaptic and control activity-dependent plasticity (*Rohrbough et al., 2000*). The only known integrin ligand at the fly NMJ is Laminin A, which is secreted from the muscle and signals through presynaptic αPS3/βν and Focal adhesion kinase 56 (Fak56) to negatively regulate the activity-dependent NMJ growth (*Tsai et al., 2012*). Teneurins have RGD motifs, but their receptor specificities remain unknown

(*Mosca et al., 2012*). Mtg secreted from the motor neurons influences postsynaptic βPS accumulation (*Rushton et al., 2009*), but that may be indirectly due to an essential role for Mtg in the organization of the synaptic cleft and the formation of the postsynaptic fields (*Rohrbough et al., 2007*; *Rushton et al., 2012*). The large size of these proteins and the complexity of ECM-integrin interactions made it difficult to recognize relevant ligand-receptor units and genetically dissect their roles in synapse development.

Here, we report the functional analysis of Tenectin (Tnc), an integrin ligand secreted from both motor neurons and muscles; Tnc accumulates at synaptic terminals and functions in *cis* to differentially engage presynaptic and postsynaptic integrin. We uncovered *tnc*, which encodes a developmentally regulated RGD-containing integrin ligand (*Fraichard et al., 2006*; *Fraichard et al., 2010*), in a screen for ECM candidates that interact genetically with *neto*, a gene essential for NMJ assembly and function (*Kim et al., 2012*). We found that Tnc selectively recruits the αPS2/βPS integrin at synaptic locations, without affecting integrin anchoring at muscle attachment sites. Dissection of Tnc functions revealed pre- and postsynaptic biologically active *cis* Tnc/integrin complexes that function to regulate neurotransmitter release and postsynaptic architecture. Finally, we exploited the remarkable features of this selective integrin ligand to uncover a novel synaptic function for integrin, in engaging the spectrin-based membrane skeleton.

## Results

### Tnc localizes at synaptic terminals

To search for novel ECM proteins important for NMJ development we set up a synthetic lethality screen that took advantage of the 50% lethality of an allele with suboptimal levels of Neto, $neto^{109}$ (*Kim et al., 2012*). Neto, an obligatory subunit of ionotropic glutamate receptor (iGluR) complexes, controls the distribution and function of iGluRs as well as the assembly and organization of postsynaptic structures (*Han et al., 2015*; *Kim et al., 2015*; *Ramos et al., 2015*). Using this lethality screen we have previously uncovered genetic interactions between *neto* and several BMP pathway components (*Sulkowski et al., 2014*; *Sulkowski et al., 2016*). Lowering the dose of Mtg, an ECM protein known to organize the synaptic cleft (*Rohrbough et al., 2007*), induced 95% lethality (n = 286) in $neto^{109}/Y;; mtg^1/+$ animals, further validating our strategy. We focused on ECM candidates (*Broadie et al., 2011*) and identified a set of overlapping deficiencies (Df(3R)BSC-318,,−492, −494, and −655) that drastically increased the lethality of $neto^{109}$ hemizygotes (from 50% for $neto^{109}/Y$ up to 82% for $neto^{109}/Y;; Df/+$). Among the common loci disrupted by these deficiencies was *tnc*, a gene coding for a large mucin-type protein conserved in many insects but with no obvious mammalian homologue (*Figure 1A–B*) (*Fraichard et al., 2006*; *Syed et al., 2008*; *Fraichard et al., 2010*; *Syed et al., 2012*).

Tnc is a secreted molecule with five vWFC (von Willebrand factor type-C) protein interaction domains separated by two PTS-rich regions. In mucins, the PTS domains are highly O-glycosylated and form gel-like structures. Tnc has one RGD and several more RGD-like motifs that have been implicated in interaction with integrin (*Fraichard et al., 2010*). During development, Tnc is secreted in the lumen of several epithelial organs, including foregut, hindgut and trachea. Tnc is also expressed in the embryonic CNS. Using a polyclonal anti-Tnc antibody (Materials and methods) we found that Tnc signals are strongly enriched in the neuropile, in the proximity of the anti-Fasciclin II (FasII) positive axons (*Figure 1—figure supplement 1A,B* and [*Fraichard et al., 2006*]). The Tnc signals were absent in the CNS of a *tnc* mutant ($tnc^{EP}$- P[EPgy2]EY03355), predicted to disrupt both known *tnc* transcripts (*Syed et al., 2012*). These mutant embryos had a normal FasII pattern indicating no obvious CNS defects during late embryogenesis (*Figure 1—figure supplement 1C–D*). The $tnc^{EP}$ mutant showed partial lethality (11.6% of expected homozygous progenies were viable, n = 303), which was not enhanced in heteroallelic combinations (12.2% viability for $tnc^{EP/Df}$, n = 392) suggesting that this mutant is equivalent to a genetic null. We also generated a small deletion mutant ($tnc^{82}$) by FRT-induced recombination. These animals die as homozygous pharate adults (100% lethality, n = 221) but produce some heteroallelic escapers (17.5% viability for $tnc^{82/Df}$, n = 389).

Western blot analysis revealed a Tnc-positive band of ~300 kD (the calculated MW for Tnc is 299 kD) in extracts from brains and body-wall muscles of control larvae (*Figure 1C*). This band was

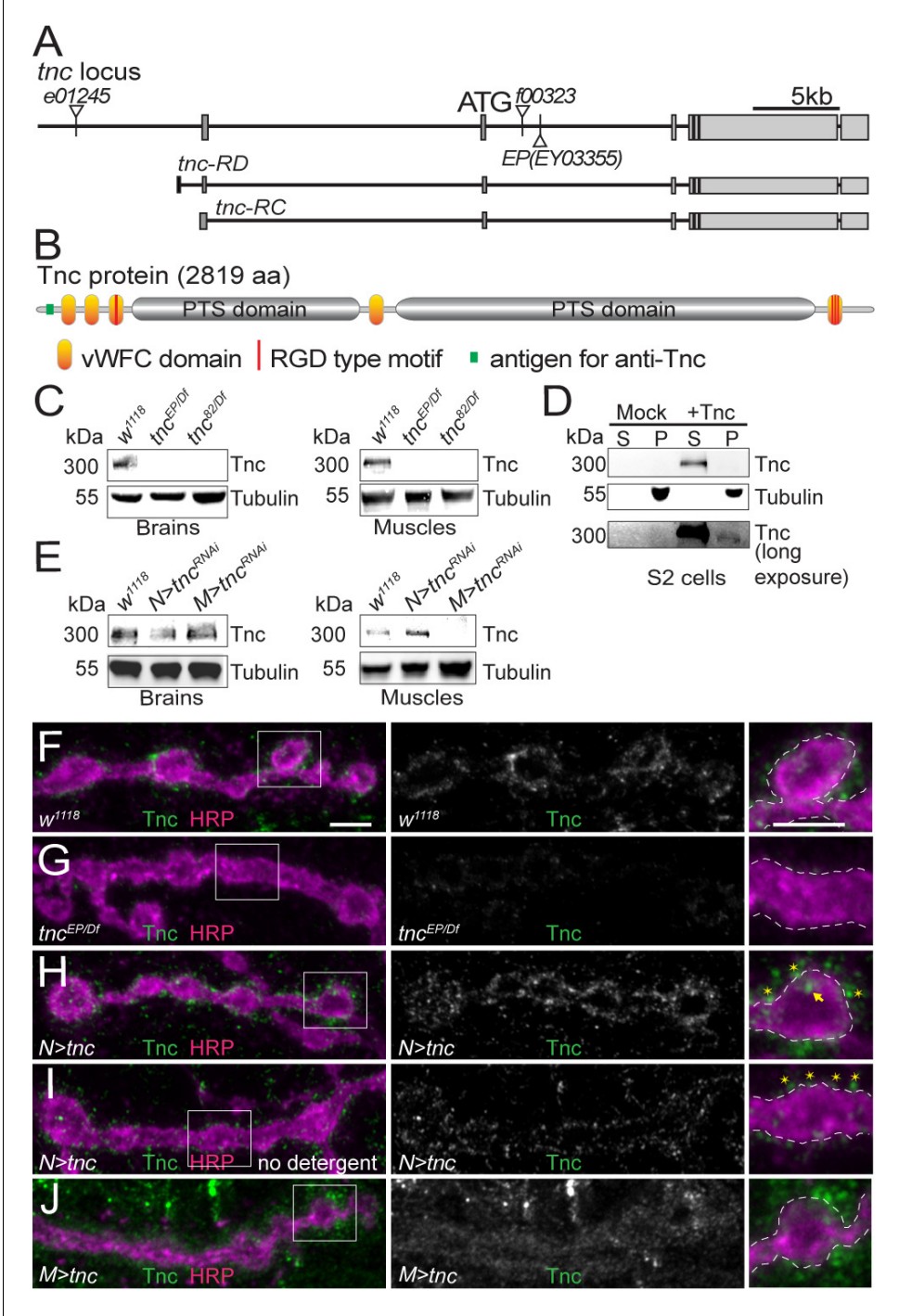

**Figure 1.** Tnc is expressed in neurons and muscles and concentrates at the synaptic cleft. (**A–B**) Diagram of the *tnc* gene and the Tnc protein domains: vWFC (orange), Pro/Thr/Ser-rich, mucin specific domains (gray), and RGD motifs (red). The antigen for the anti-Tnc antibody is marked in green. (**C–E**) Western blot analyses of lysates from larval brains or muscles and transiently transfected S2 cells. Tnc can be detected in control but not in *tnc* mutants (**C**) and is reduced by knockdown of *tnc* in neurons or muscle (**E**). Tnc is efficiently secreted in the S2 cell media (**S**) compared with the cell pellet (**P**). (**F–J**) Confocal images of NMJ4 boutons of indicated genotypes stained for Tnc (green) and HRP (magenta). Low levels of Tnc surround synaptic boutons in control but not *tnc* mutant NMJs (**F–G**). Expression of Tnc in neurons but not in the muscles induces accumulation of Tnc-positive puncta in and around the NMJ boutons. The small puncta (asterisks) appear to be extracellular, as they are still present in detergent-free staining conditions (**I**), whereas the large aggregates (arrows) likely correspond to intracellular

*Figure 1 continued on next page*

*Figure 1 continued*

secretory compartments. Scale bars: 5 µm. Genotypes: *tnc$^{EP(or\ 82)/Df}$* (*tnc$^{EP(or\ 82)}$*/Df(3R)BSC655); N > *tnc$^{RNAi}$* (BG380-Gal4/+;UAS-*tnc$^{RNAi}$*/+); M > *tnc$^{RNAi}$* (UAS-*tnc$^{RNAi}$*/G14-Gal4); N > *tnc* (UAS-*tnc*/+; elav-Gal4/+); M > *tnc* (UAS-*tnc*/BG487-Gal4).

DOI: https://doi.org/10.7554/eLife.35518.003

The following figure supplements are available for figure 1:

**Figure supplement 1.** Tnc distribution in embryonic and larval tissues.

DOI: https://doi.org/10.7554/eLife.35518.004

**Figure supplement 2.** Addition of an HA tag does not change the distribution of overexpressed Tnc.

DOI: https://doi.org/10.7554/eLife.35518.005

undetectable in *tnc* hetero-allelic combinations, *tnc$^{EP/Df}$* and *tnc$^{82/Df}$*. A band of similar size was found in S2 cells transfected with a Tnc expression construct and was enriched in the conditioned media, indicating that Tnc is efficiently secreted in cell culture (Figure 1D, Material and methods). Neuron specific RNAi knockdown reduced Tnc levels in larval brains to 43% of the control group; this generated very strong phenotypes (below) suggesting that the residual band could reflect additional Tnc-expressing cells in the larval brain. The muscle-specific knockdown reduced the muscle Tnc levels to 19% of the control (*Figure 1E*).

During larval stages, we found Tnc positive signals throughout the muscles with weak accumulation at the NMJ (*Figure 1F* and *Figure 1—figure supplement 1F*). Under the same imaging conditions, the signals were significantly reduced in *tnc$^{EP/Df}$* mutants (*Figure 1G* and *Figure 1—figure supplement 1G*, quantified in *Figure 1—figure supplement 1E*). Such weak NMJ immunoreactivities were previously reported for proteins secreted in the synaptic cleft (*Rushton et al., 2009*). To further confirm the specificity of Tnc signals, we overexpressed Tnc in motor neurons (*elav-Gal4*) or muscles (*BG487-Gal4*) and examined the NMJs (*Figure 1H–J* and *Figure 1—figure supplement 1H–I*). Paneuronal expression of Tnc induced strong accumulation of Tnc-positive signals at synaptic terminals, as well as along the motor neuron axons. At these NMJs, Tnc-labeled puncta were concentrated at the edge of anti-horseradish peroxidase (HRP)-stained boutons (*Jan and Jan, 1982*). Most of these signals were also observed in the absence of detergents, suggesting that Tnc is secreted in the synaptic terminal. In contrast, muscle overexpression of *tnc* showed increased Tnc-positive signals throughout the muscle; at synaptic terminals these signals appeared diffuse and farther away from the neuronal membrane. Thus, excess muscle Tnc may not be properly targeted and/or stabilized at synaptic terminals and may have detrimental effects on Tnc-mediated functions. We repeated these results using independent HA-tagged *tnc* transgenes and staining with anti-HA antibodies (*Figure 1—figure supplement 2*). Both tagged and untagged transgenes rescued the viability of *tnc* mutants (see below), indicating that Tnc functions are unaffected by addition of the tag. As above, expression of *tnc-HA* in neurons but not in muscles induced high accumulation of HA-positive puncta, accessible without detergent, around the synaptic boutons, indicating extracellular distribution (*Figure 1—figure supplement 2A–D*). Thus, Tnc is expressed in both motor neurons and muscles and appears to accumulate at the ECM surrounding synaptic terminals.

## *tnc* mutants have impaired NMJ physiology

To investigate a possible role for Tnc in the function of the nervous system and/or the musculature, we examined the morphology and physiology of *tnc* mutants. During larval stages, both *tnc* mutants (*tnc$^{EP/Df}$* and *tnc$^{82/Df}$*) had largely normal NMJ, with minimally increased bouton numbers (*Figure 2—figure supplement 1A–B*). A closer examination revealed smaller boutons, with less clear bouton-interbouton delimitations (details below). The few adult escapers did not fly and exhibited climbing defects (*Figure 2—figure supplement 1C*). Such phenotypes are consistent with previously reported flightless adults generated by RNAi-mediated Tnc knockdown (*Fraichard et al., 2010*).

We next recorded the evoked excitatory junction potentials (EJPs) and spontaneous miniature excitatory junction potentials (mEJPs) from muscle 6 of third instar larvae (*Figure 2A–E*). The mEJPs amplitude was normal in *tnc* mutants. However, the mean frequency of mEJPs was significantly reduced in *tnc* mutants compared with the control (*w$^{1118}$*, 2.67 ± 0.14 Hz vs. *tnc$^{EP/Df}$*, 1.68 ± 0.13 Hz, p=0.0001, and *tnc$^{82/Df}$*, 2.11 ± 0.14 Hz, p=0.0322, *Figure 2D*). *tnc* mutations caused 23% and 18% reduction in evoked EJPs amplitude of *tnc$^{EP/Df}$* and *tnc$^{82/Df}$* animals, respectively (*w$^{1118}$*, 26.30 ± 0.99

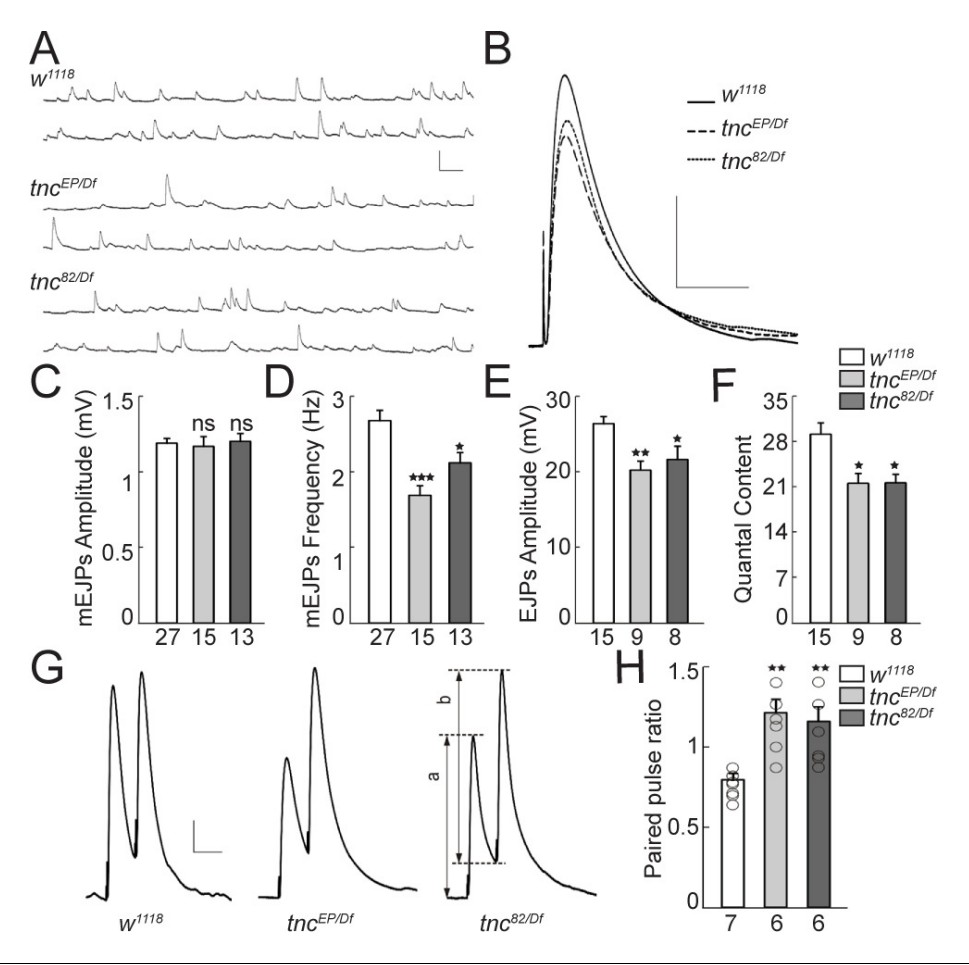

**Figure 2.** Reduced vesicle release probability at *tnc* mutant NMJs. (A–B) Representative traces of spontaneous (A) and evoked (B) neurotransmitter release recorded from muscle 6 of indicated genotypes at 0.5 mM $Ca^{2+}$. (C–F) Summary bar graphs showing the mean amplitude (C) and frequency (D) of mEJPs, the mean amplitude of EJPs (E) and the quantal content (F). The mEJPs amplitude is normal in *tnc* mutants, but the mEJPs frequency, EJPs amplitude and quantal content are reduced in both *tnc* allelic combinations. Resting potential: $w^{1118}$ −61.30 ± 0.26 mV, $tnc^{EP/Df}$ −61.27 ± 0.38 mV, $tnc^{82/Df}$ −61.85 ± 0.97 mV; input resistance: $w^{1118}$ 7.39 ± 0.40 MΩ, $tnc^{EP/Df}$ 7.52 ± 0.72 MΩ, $tnc^{82/Df}$ 6.74 ± 0.24 MΩ. (G) Representative traces for paired-pulse stimulation in larvae of indicated genotypes. Paired stimuli (200 μsec, 1.9 V) were separated by duration of 50 ms. (H) Quantification of the paired-pulse ratio (% change in the amplitude of the second EJP (b) to that of the first EJP (a)) in larvae of indicated genotypes. The number of NMJs examined is indicated under each bar. Bars indicate mean ± SEM. **p<0.01, *p<0.05. Scale bars: (A): 1.5 mV- 500 ms; (B): 10 mV- 400 ms; (G): 4 mV- 50 ms.

DOI: https://doi.org/10.7554/eLife.35518.006

The following figure supplement is available for figure 2:

**Figure supplement 1.**

DOI: https://doi.org/10.7554/eLife.35518.007

mV vs. $tnc^{EP/Df}$, 20.15 ± 1.26 mV, p=0.0042, and $tnc^{82/Df}$, 21.53 ± 1.38 mV, p=0.0375, *Figure 2E*). Moreover, *tnc* mutants showed a significant decrease in quantal content ($w^{1118}$, 29.05 ± 1.81 vs. $tnc^{EP/Df}$, 21.44 ± 1.60 and $tnc^{82/Df}$, 21.53 ± 1.38, *Figure 2F*). Since we found no change in the resting potential and input resistance in mutant animals, the decrease in EJPs amplitude and quantal content was probably not caused by abnormal passive membrane properties in the muscle. Instead, the reduction of quantal content could be due to a decreased number of vesicle release sites or reduced probability of release. The reduced mEJP frequency is in agreement with reduced quantal content due to fewer release sites. To evaluate the vesicle release probability, we measured the paired-pulse

ratio (PPR) using the EJP amplitudes evoked by two stimuli separated by duration of 50 ms (*Wong et al., 2014*). At 0.5 mM extracellular $Ca^{2+}$ concentration, the control larvae showed mild short-term depression following paired-pulse stimulation (PPR < 1, *Figure 2G–H*), indicating a relatively high initial probability of vesicle release; the second stimulus, provided before the resting $Ca^{2+}$ returned to baseline, lead to the exocytosis of fewer synaptic vesicles than the first stimulus. In contrast, the *tnc* mutant NMJs showed elevated facilitation and significantly increased PPR ($w^{1118}$, 0.80 ± 0.03 vs. $tnc^{EP/Df}$, 1.21 ± 0.08, p=0.0231, and $tnc^{82/Df}$, 1.16 ± 0.09, p=*0.0295*). The higher the ratio of EJP amplitudes following the first and second pulses, the lower is the probability of release. Thus, *tnc* mutants have significantly decreased probability of vesicle release.

## Neuronal Tnc modulates neurotransmitter release

Since Tnc is expressed in both pre- and post-synaptic compartments, we next examined which Tnc pool(s) is required for NMJ function. We found that *tnc* knockdown in motor neurons resulted in mEJPs with significantly reduced frequency as compared to the controls ($tnc^{RNAi}$ transgene with no driver, or driver alone) (control, 2.69 ± 0.2 Hz vs. $N > tnc^{RNAi}$, 1.31 ± 0.10 Hz p=0.0001, and $M > tnc^{RNAi}$, 2.4 ± 0.17 Hz p=0.4434) (*Figure 3A–C*, and *Figure 3—figure supplement 1*). Similar results were obtained with a second RNAi line (*GD14952*) confirming that these phenotypes are specific to *tnc* depletion (*Figure 3—figure supplement 1*). In contrast, *tnc* knockdown in muscles had no effect on mEJPs frequency or amplitude. Thus, neuronal but not muscle Tnc is required for normal neurotransmitter release.

Expression of Tnc in neurons but not in muscles also rescued the mEJPs frequency and EJPs amplitude defects observed in *tnc* mutants (*Figure 3D–F*). Similar to *tnc* mutants, mutants carrying only a *tnc* transgene but no driver showed reduced mEJP frequency and EJP amplitude (compare *Figure 3D–F* and *Figure 2*). Expression of the *tnc* transgene in neurons, but not in muscles of *tnc* mutants restored the mEJPs frequency ($w^{1118}$, 1.97 ± 0.27 Hz vs. N > tnc; $tnc^{EP/Df}$, 1.56 ± 0.18 Hz p=0.07 and M > tnc; $tnc^{EP/Df}$, 1.20 ± 0.16 Hz p=0.0019) and EJPs amplitude ($w^{1118}$, 71.51 ± 2.39 mV vs. $N > tnc$; $tnc^{EP/Df}$, 70.1 ± 2.08 mV p=0.198 and M > tnc; $tnc^{EP/Df}$, 45.03 ± 3.67 mV p<0.0001) to levels that are no longer significantly different from the $w^{1118}$ control. Together, these data indicate that neuron-derived Tnc regulates normal neurotransmitter release at the NMJ.

Previous studies suggest that Tnc functions as a ligand for αPS2/βPS integrin during wing morphogenesis (*Fraichard et al., 2010*). *Drosophila* integrins have been implicated in NMJ growth and synaptic function (*Keshishian et al., 1996*), with the βPS-containing complexes primarily in the postsynaptic compartment (*Prokop et al., 1998*; *Beumer et al., 1999*; *Koper et al., 2012*). If Tnc functions by recruiting integrin at the NMJ, then presynaptic but not postsynaptic βPS or αPS2 should similarly modulate the neurotransmitter release. Indeed, neuronal knockdown of *myospheroid* (*mys*), which encodes the βPS integrin subunit, significantly reduced the mEJPs frequency compared to control ($mys^{RNAi}$ transgene with no driver) (*Figure 3G–H*). In contrast, muscle knockdown of *mys* had no detectable effect on mEJPs frequency (control, 0.95 ± 0.14 Hz vs. $N > mys^{RNAi}$, 0.47 ± 0.05 Hz p=0.0044 and $M > mys^{RNAi}$, 1.00 ± 0.07 Hz p=0.9142) or amplitude. Similarly, neuronal but not muscle knockdown of *inflated* (*if*), which codes for αPS2 integrin, significantly reduced the mEJPs frequency compared to control ($if^{RNAi}$ transgene with no driver) (*Figure 3J–L*) (control, 1.62 ± 0.51 Hz vs. $N > if^{RNAi}$, 0.99 ± 0.23 Hz p=0.0041 and $M > if^{RNAi}$, 1.67 ± 0.42 Hz p=0.9662). The $if^{RNAi}$ transgene appeared stronger than $mys^{RNAi}$ and generated larval lethality and, occasionally, muscle attachment defects; using these transgenes we observed diminished mEJP amplitude compared to control (*Figure 3L*). Overall, the reduction of presynaptic αPS2/βPS mirrored the mEJP frequency deficits observed for *tnc* neuronal knockdown suggesting that neuron-derived Tnc functions as a ligand for presynaptic αPS2/βPS to modulate neurotransmitter release. Postsynaptic αPS2/βPS integrin does not appear to influence neurotransmitter release. Alternatively, a role for postsynaptic αPS2/βPS may be obscured by partial knockdowns and essential functions for these genes in the muscle.

If Tnc recruits integrin to modulate neurotransmitter release, then *tnc* and *mys* (or *if*) should interact genetically. We tested this prediction by examining the trans-heterozygote animals (*mys/+;; tnc/+*). Indeed, these trans-heterozygotes exhibited severe mEJP deficits that resembled *tnc* mutants, whereas individual heterozygote larvae (*mys/+* or *tnc/+*) showed normal mEJPs frequency and amplitude (*Figure 3M–O*). This indicates that *tnc* and *mys* function together to modulate neurotransmitter release.

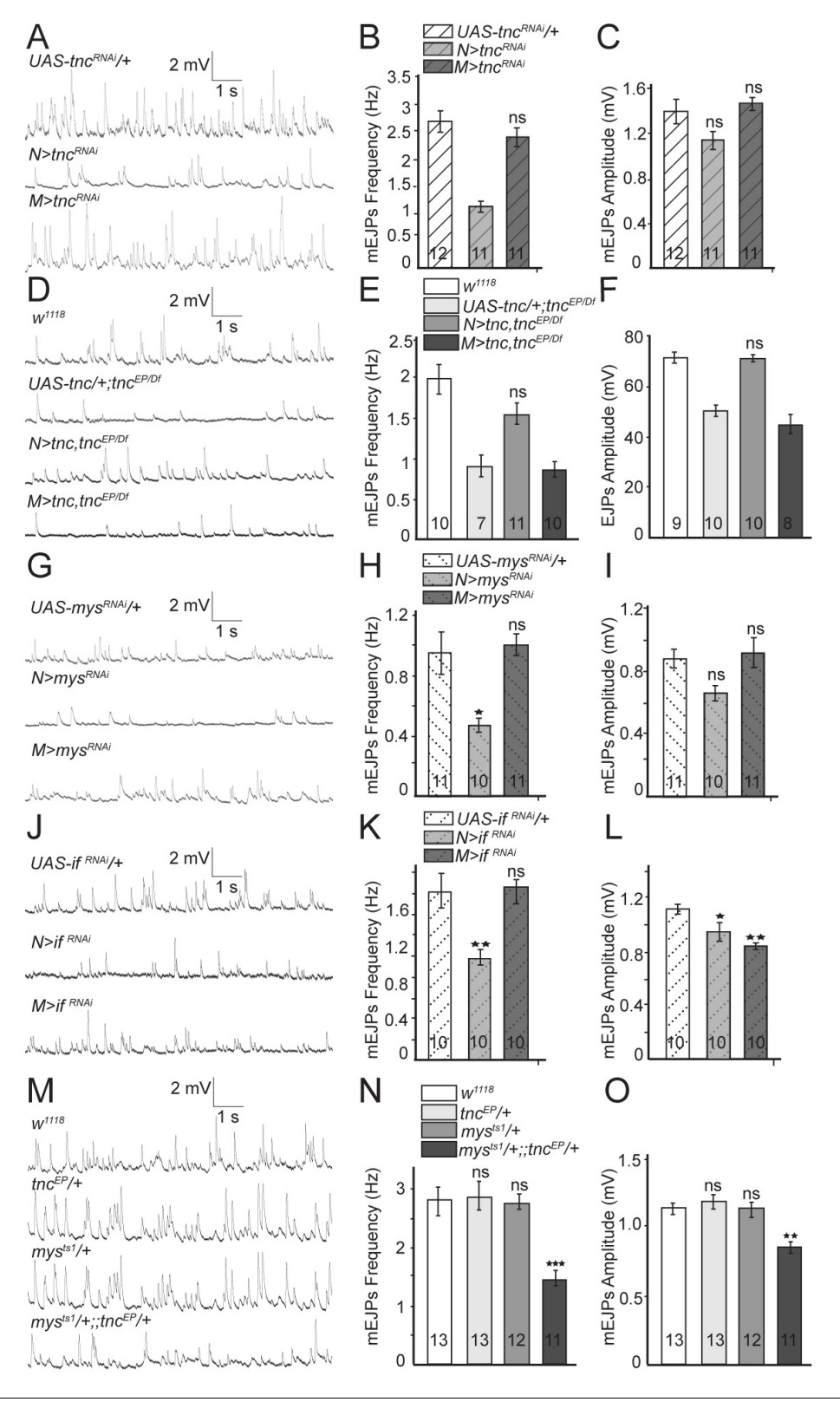

**Figure 3.** Presynaptic Tnc and integrin are critical for neurotransmitter release. (**A–L**) Representative traces and summary bar graph for mEJPs and EJPs recorded at 0.8 mM Ca$^{2+}$ from muscle 6 of indicated genotypes. The number of samples examined is indicated in each bar. (**A–C**) Neuronal knockdown of *tnc* significantly reduces the mEJPs frequency. (**D–F**) Neuronal but not muscle expression of Tnc can rescue the mEJPs frequency and EJP

*Figure 3 continued on next page*

*Figure 3 continued*

amplitude at *tnc* mutant NMJs. (**G–L**) The mean mEJPs frequency is dramatically reduced when *mys/βPS integrin* or *if/αPS2* are knocked down in the neurons. Knockdown of *if/αPS2* also induces slight reduction of the mean mEJPs amplitude and occasionally muscle attachment defects. (**M–O**) The trans-heterozygotes (*mys/+;; tnc/+*) show enhancement of phenotypes compared with individual heterozygotes, indicating that *tnc* and *mys* interact genetically. Bars indicate mean ±SEM. ns, not significant (p>0.05), ***p<0.001,***p<0.001 *p<0.05. Genotypes: N > tnc^RNAi (BG380-Gal4/+; UAS-tnc^RNAi/+); M > tnc^RNAi (UAS-tnc^RNAi/+; 24B-Gal4/+); *tnc* rescue control (UAS-tnc/+;tnc^EP/Df(3R)BSC655); N > tnc, tnc^EP/Df (UAS-tnc/+;tnc^EP/elav-Gal4,Df(3R)BSC655); M > tnc, tnc^EP/Df (UAS-tnc/+; tnc^EP/24B-Gal4, Df(3R)BSC655); N > mys^RNAi (BG380-Gal4/+; UAS-Dcr-2/+; UAS-mys^RNAi/+); M > mys^RNAi (UAS-Dcr-2/+; UAS-mys^RNAi/24B-Gal4); N > if ^RNAi (BG380-Gal4/+; UAS-if ^RNAi/UAS-Dcr-2); M > if ^RNAi (UAS-if ^RNAi/UAS-Dcr-2; 24B-Gal4/+).

DOI: https://doi.org/10.7554/eLife.35518.008

The following figure supplement is available for figure 3:

**Figure supplement 1.** Additional control recordings.

DOI: https://doi.org/10.7554/eLife.35518.009

## Muscle Tnc recruits postsynaptic integrin

If Tnc recruits and/or stabilizes βPS integrin at synaptic terminals, then Tnc should co-localize with βPS and form Tnc/integrin complexes at synaptic locations and perturbations of Tnc should alter the recruitment of βPS integrin at larval NMJ. Indeed, the βPS signals were dramatically reduced at *tnc* mutant NMJs (*Figure 4A–D*). In these analyses the βPS immunoreactivities concentrated at perisynaptic locations, surrounding the control boutons, consistent with previous observations that the muscle pool constitutes the major fraction of βPS at synaptic terminals (*Beumer et al., 1999*). Interestingly, the βPS levels remained unchanged at the muscle attachment sites; we also observed no detachment of the muscle fibers or defects in costamere organization (*Maartens and Brown, 2015*). This indicates that loss of Tnc selectively impairs the recruitment of βPS integrin at synaptic terminals. The anti-Tnc antibodies marked discrete puncta at the edge of the HRP- labeled boutons in a region strongly stained by anti-βPS antibodies, but Tnc was not detectable at the muscle attachment sites (*Figure 4E–F*). This suggests that Tnc and βPS may directly associate at synaptic terminals. Our attempts to co-immunoprecipitate Tnc/integrin complexes from larval carcasses failed; instead, we tested for their close juxtaposition at the NMJ using a proximity ligation assay (PLA), which indicates a less than 40 nm distance between two proteins (*Wang et al., 2015*). As shown in *Figure 4G–H*, PLA signals between Tnc and βPS were detected at control but not at *tnc* mutant NMJs. These PLA signals were tightly packed around the synaptic boutons, unlike the Tnc or βPS immunoreactivities, which spread into the postsynaptic specializations, suggesting that Tnc and βPS form complexes in the close proximity of the synaptic terminal. Such complexes could influence the presynaptic neurotransmitter release, as described above, but could also function in the postsynaptic compartment, where most of the perisynaptic βPS resides (see below).

Since different integrin heterodimers provide spatial and temporal specificities, we next examined the distribution of integrin α-subunit(s) at *tnc* boutons. We found that αPS2 but not αPS1 levels were selectively reduced at *tnc* mutant NMJs (*Figure 4I–K* and *Figure 4—figure supplement 1A–B*), consistent with the RGD-containing Tnc being a ligand for αPS2/βPS (*Fraichard et al., 2010*). In addition, the levels of phosphorylated Fak were normal at *tnc* mutant NMJs (*Figure 4—figure supplement 1C–D*) suggesting that Tnc does not influence the LanA-induced αPS3/βν activation of the Fak signaling pathway (*Tsai et al., 2008*; *Tsai et al., 2012*). The synaptic abundance of βPS integrin subunit was inversely correlated with the synaptic accumulation of FasII, a homophilic adhesion molecule required for synapse stabilization and growth (*Schuster et al., 1996a*; *Schuster et al., 1996b*; *Beumer et al., 2002*). We found that FasII synaptic levels were increased by 40% (p<0.05, n = 28) in *tnc* mutants compared with the controls (*Figure 4—figure supplement 1E–F*). This increase resembles the elevated levels of FasII reported at *mys* mutant NMJs, and indicates that loss of Tnc recapitulates some of the phenotypes reported for selective *mys* mutants (*Beumer et al., 2002*).

Since secreted Tnc accumulates at synaptic terminals, both neuron- and muscle-derived Tnc could potentially recruit βPS. However, we found that *tnc* knockdown in muscles, but not in neurons, reduced the synaptic βPS levels (*Figure 4M–O*, quantified in P). In fact, *tnc* knockdown in neurons induced a significant increase of βPS synaptic levels, suggesting that neuron-derived Tnc limits the

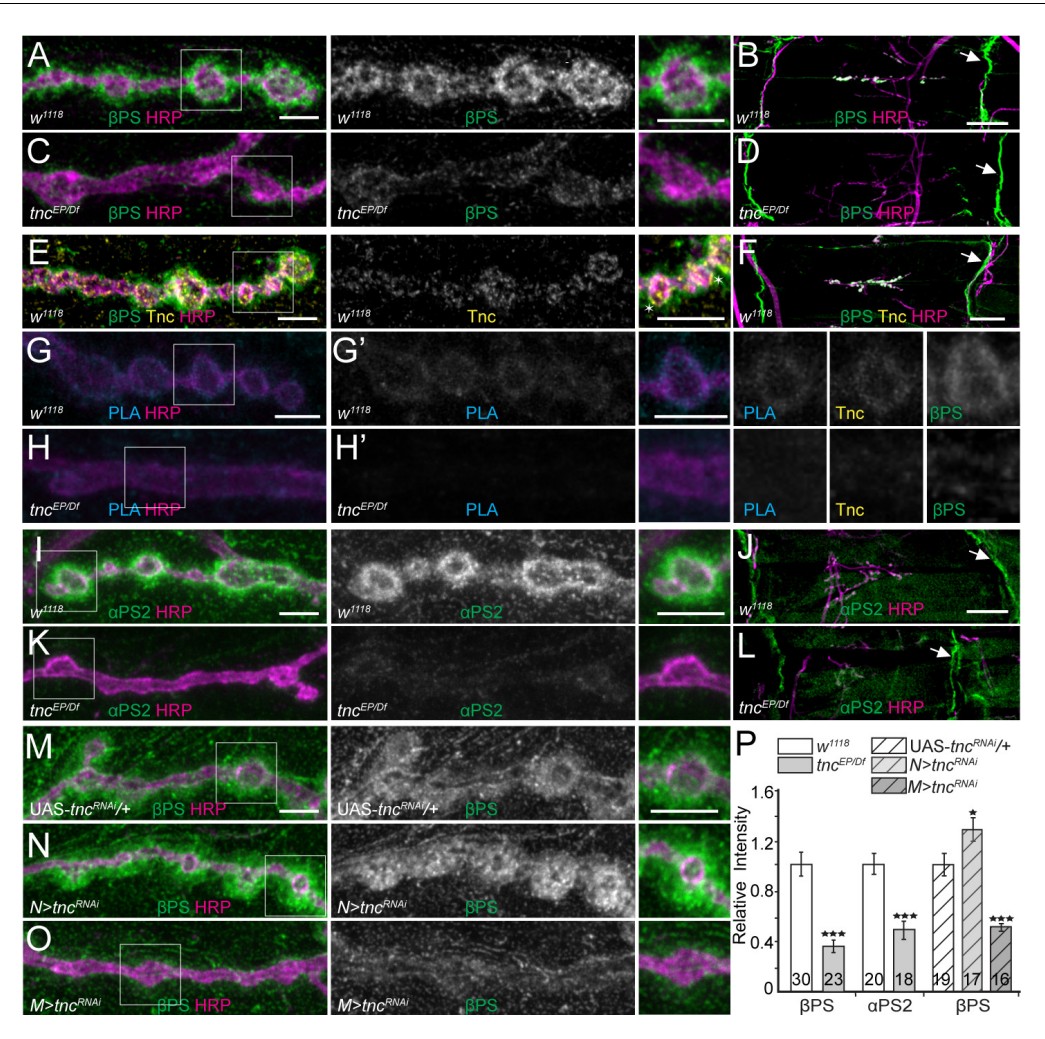

**Figure 4.** *tnc* mutants have reduced perisynaptic αPS2/βPS integrin. (A–D) Confocal images of control NMJ4 boutons and NMJ 6/7 muscle fields of indicated genotypes stained for βPS (green) and HRP (magenta). Compared to control (*w^1118*), *tnc* mutant have dramatically decreased βPS signals at the NMJs (quantified in P), but normal levels at the muscle attachment sites (arrows). (E–F) Confocal images of control NMJ4 boutons stained with Tnc (yellow), βPS (green) and HRP (magenta). Like βPS, Tnc concentrates at the periphery of HRP-marked boutons (asterisks), but unlike βPS, Tnc is not present at the muscle attachment sites. (G–H') Distribution of Tnc (yellow), βPS (green), PLA signals (cyan) (G'–H'), and HRP (magenta) in control and *tnc* mutant boutons. PLA signals are only observed at control NMJs and localize circumferentially to the boutons, indicating that Tnc and βPS are in close proximity at the synaptic cleft. (I–P) Confocal images of NMJ4 boutons and NMJ6/7 muscle fields of indicated genotypes stained for αPS2 (I–L) or βPS (M–O) (green), and HRP (magenta). Similar to βPS, αPS2 signals are dramatically reduced at *tnc* mutant NMJs (quantified in P), but are normal at the muscle attachment sites (arrows). Tissue specific *tnc* knockdown indicates that postsynaptic Tnc controls the βPS accumulation at synaptic locations, whereas neuron-derived Tnc appears to limit it (M–O). The number of NMJs examined is indicated in each bar. Bars indicate mean ± SEM. ***p<0.001, *p<0.05. Scale bars: (A, E, G, G', I and M) 5 μm; (B, F and J) 20 μm. Genotypes: *tnc^EP/Df* (*tnc^EP/Df(3R)BSC655*); N > *tnc^RNAi* (*BG380-Gal4/+; UAS-tnc^RNAi/+*); M > *tnc^RNAi* (*UAS-tnc^RNAi/+; 24B-Gal4/+*)..

DOI: https://doi.org/10.7554/eLife.35518.010

The following figure supplements are available for figure 4:

**Figure supplement 1.** Comparison of various synaptic proteins in control and tnc mutant NMJs.
DOI: https://doi.org/10.7554/eLife.35518.011

**Figure supplement 2.** The inhibitory effect of neuron-derived Tnc on muscle Tnc.
DOI: https://doi.org/10.7554/eLife.35518.012

accumulation of predominantly postsynaptic βPS. This unexpected result prompted us to examine the distribution of Tnc itself in RNAi experiments (*Figure 4—figure supplement 2*). Compared to the control (*tnc^RNAi* transgene with no driver), *tnc* knockdown in motor neurons produced a significant increase (by 28%, p=0.0044, n = 27) of synaptic Tnc levels; this result is consistent with the apparent increase in Tnc net levels in muscle extracts from *N > tnc^RNAi* larvae (*Figure 1E*, right panel). Thus, neuron-derived Tnc limits the accumulation of muscle-derived Tnc at synaptic terminals. In contrast, *tnc* knockdown in muscle diminished the synaptic Tnc levels by 37% (p<0.0001, n = 30). This partial reduction may reflect an inefficient RNAi treatment and/or a complementary increase in the neuron-derived Tnc. Nonetheless our data indicate that muscle Tnc is required in cis for the postsynaptic recruitment of βPS and that neuron-derived Tnc limits the accumulation of both Tnc and βPS postsynaptically (see below).

To determine the function of Tnc in the muscle we first tested whether Tnc influences the assembly and organization of postsynaptic iGluR fields by examining the levels and distribution of various postsynaptic components. *Drosophila* NMJ utilizes two types of iGluRs, type-A and -B, which require the essential auxiliary protein Neto for their distribution and function. Lack of Tnc did not alter the intensities of GluRIIA and GluRIIB synaptic signals or the IIA/IIB ratio (*Figure 5—figure supplement 1*). This result is consistent with the normal mini amplitude observed at *tnc* mutant NMJs (*Figure 2*). Neto itself appeared properly recruited at Tnc-depleted synapses (*Figure 5—figure supplement 1*). In addition to iGluRs, Neto is critical for the recruitment of p21-activated kinase, PAK, a postsynaptic protein that stabilizes type-A receptors at PSDs (*Ramos et al., 2015*). We found that PAK signals are normal at Tnc-deprived NMJs, even though the βPS levels are reduced (*Figure 5—figure supplement 1*). This suggests that Neto controls PAK recruitment at synaptic terminals; alternatively, a very low level of βPS may suffice in recruiting/stabilizing PAK at synaptic terminals.

In the course of these experiments, we noted that *tnc* mutant NMJs have smaller boutons and often poorly defined bouton/interbouton boundaries. To characterize these defects, we first examined the distribution of HRP-marked neuronal membranes and Discs large (Dlg), a PDZ (PSD-95/Dlg/Zona occludens-1) domain-containing scaffolding protein (*Budnik et al., 1996*). Dlg localizes perisynaptically to the subsynaptic reticulum (SSR), a stack of membrane folds that surrounds the type I boutons (*Guan et al., 1996*). In the absence of Tnc, the type Ib boutons appeared significantly smaller and had diminished Dlg signals (*Figure 5A–B*, quantified in F-G). *tnc* knockdown in neurons induced a slight increase in the Dlg signals and no change in the bouton area, whereas *tnc* knockdown in muscles significantly decreased both the Dlg synaptic levels and the size of the type Ib boutons (*Figure 5C–E*). Similar reduction in bouton size has been reported for selective *mys* mutants (*Beumer et al., 1999*), suggesting that postsynaptic Tnc/integrin complexes control bouton size.

In electron micrographs, type Ib control boutons are surrounded by a thick SSR (*Figure 5H*, quantified in 5I); presynaptic T-bars and electron-dense membranes mark individual synapses. The synapses appeared to have normal organization at *tnc* mutant boutons, with electron-dense synaptic membranes separated by a dense synaptic cleft (*Figure 5J*). However, the mutant boutons had irregular shapes and were surrounded by sparse SSR, with wider spaces between the membrane layers. These ultrastructural defects are consistent with the observed morphological phenotypes and indicate that muscle Tnc regulates the SSR thickness and bouton architecture.

## *tnc* mutants have disrupted spectrin-based membrane skeleton

The *tnc* phenotypes may reflect increased adhesion due to elevated FasII levels (*Figure 4—figure supplement 1*). Alternatively, bouton architecture could be disrupted when muscles contract in the absence of a properly reinforced synaptic skeleton, including the microtubule-based and the cortical membrane skeleton. We found no microtubule defects at *tnc* mutant NMJs (*Figure 6A–B*). At the presynaptic arbor, the mature microtubule bundles that traverse the NMJ branches can be visualized with antibodies against the microtubule-associated protein Futsch. These microtubules remained organized in smooth sheaths and loops in *tnc* mutant larvae, similar to the control.

In contrast, loss of *tnc* severely disrupted the α-Spectrin accumulation at the NMJs (*Figure 6C–F*). The net levels of α-Spectrin were normal in extracts from *tnc* larval muscles, as determined by Western blot analysis, but the synaptic abundance of α-Spectrin at *tnc* NMJs was reduced to 37% from control levels (p<0.001, n = 21). Since α-Spectrin is essential for the integrity of the SSR (*Pielage et al., 2006*), loss of synaptic α-Spectrin is consistent with the sparse SSR observed at *tnc* mutant NMJs. Moreover, *α- and β-spectrin* mutant embryos have reduced neurotransmitter release

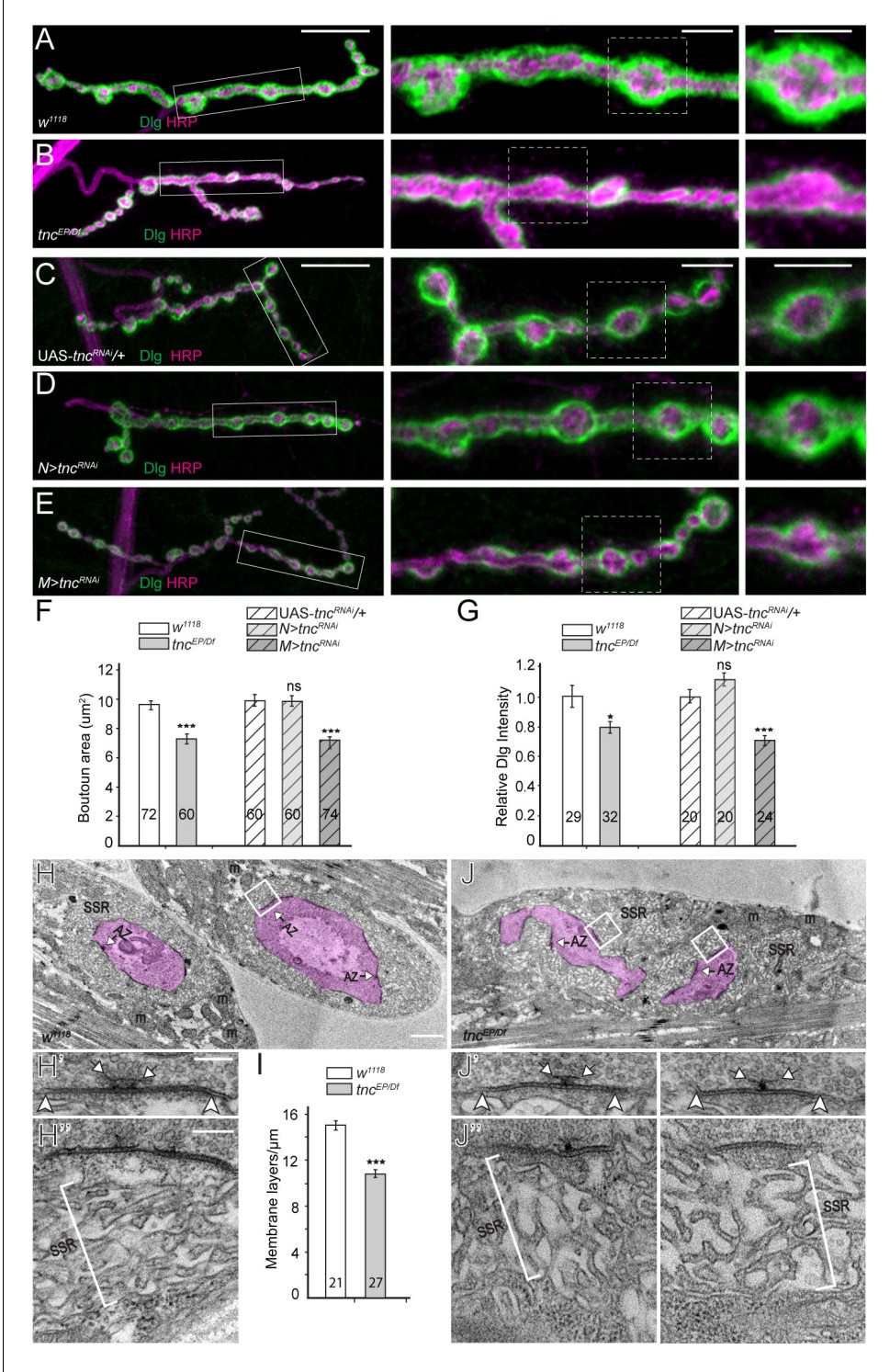

**Figure 5.** *tnc* mutants have smaller boutons and reduced SSR. (**A–E**) Confocal images and analyses of third instar NMJ4 boutons of indicated genotypes stained for Dlg (green) and HRP (magenta). Compared to control, *tnc* mutant NMJs have smaller boutons and reduced perisynaptic Dlg levels (quantified in **F–G**). Knockdown of *tnc* in muscles but not in motor neurons recapitulates the mutant defects. (**H–J**) Electron micrographs of type Ib boutons. The neuronal compartment is labeled in magenta; the active zones (AZ, arrows), mitochondria (**m**), and subsynaptic reticulum (SSR, brackets) are indicated. The *tnc* mutants have sparse SSR with reduced density of the membrane layers (quantified in I). The *tnc* synapses appear normal (insert detail) but they reside in relatively distorted boutons. The number of samples examined is indicated in each bar. Bars indicate mean ±SEM. ns

*Figure 5 continued on next page*

*Figure 5 continued*

(p>0.05), *p<0.05, ***p<0.001. Scale bars: (A) 20 µm, 5 µm in boutons; (E) 2 µm, 200 nm in details. Genotypes: N > tnc$^{RNAi}$ (BG380-Gal4/+; UAS-tnc$^{RNAi}$/+); M > tnc$^{RNAi}$ (UAS-tnc$^{RNAi}$/+; 24B-Gal4/+)..

DOI: https://doi.org/10.7554/eLife.35518.013

The following figure supplement is available for figure 5:

**Figure supplement 1.** tnc mutants have normal PSDs.

DOI: https://doi.org/10.7554/eLife.35518.014

and diminished EJPs amplitudes without any apparent defects in postsynaptic receptor fields (*Featherstone et al., 2001*); these defects are reminiscent of Tnc-deprived NMJs. Interestingly, the distorted shapes of *tnc* mutant boutons resemble the less individuated boutons seen in *α-spec-trin$^{R22S}$* larvae, which are impaired for spectrin tetramerization (*Khanna et al., 2015*). Tetrameriza-tion is required for formation of the spectrin-based membrane skeleton (SBMS) but is not required for viability in *Drosophila*, whereas *spectrins* are essential genes (*Lee et al., 1993*; *Pielage et al., 2005*; *Pielage et al., 2006*). A key protein involved in the organization of SBMS is Adducin (*Bennett and Baines, 2001*). *Drosophila adducin* gene encodes several isoforms, all but one

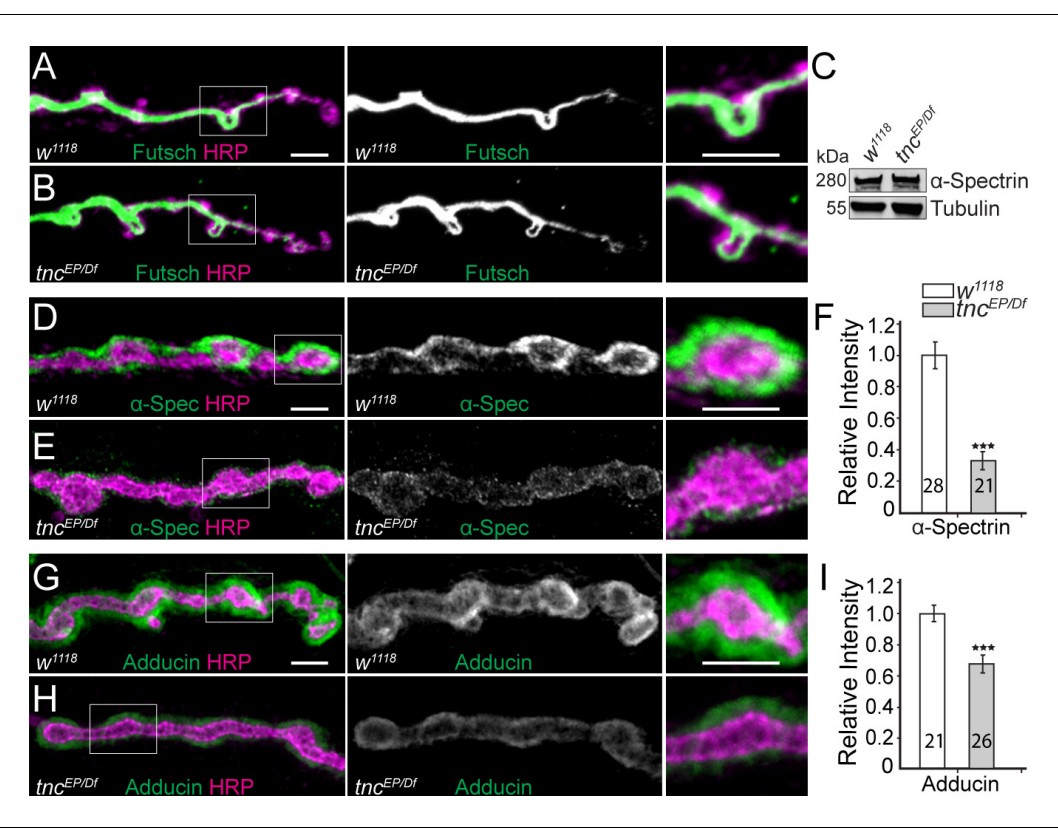

**Figure 6.** Diminished cortical skeleton at *tnc* mutant NMJs. (**A–B**) Confocal images of NMJ4 boutons stained for Futsch (green) and HRP (magenta) reveal normal presynaptic Futsch-positive loops and microtubules bundles at *tnc* mutant NMJs. (**C**) Western blot analysis of lysates from larval carcasses show normal levels of α-Spectrin in *tnc* mutants. (**D–I**) Confocal images of NMJ4 boutons for the indicated genotypes stained for α-Spectrin (**D–E**), or Adducin (**G–H**) (green) and HRP (magenta), (quantified in F and I). α-Spectrin levels are dramatically decreased at *tnc* mutant NMJs; the reduction of Adducin is less drastic, but significant. The number of NMJs examined is indicated in each bar. Bars indicate mean ± SEM. ns (p>0.05), ***p<0.001. Scale bars: 5 µm.

DOI: https://doi.org/10.7554/eLife.35518.015

The following figure supplement is available for figure 6:

**Figure supplement 1.** Tnc/integrin-mediated spectrin recruitment.

DOI: https://doi.org/10.7554/eLife.35518.016

detectable with the anti-Adducin antibody 1B1 (*Wang et al., 2014*). Using this antibody, we found that Adducin is also diminished at *tnc* mutant terminals (*Figure 6G–I*). These data suggest that Tnc is required for the proper recruitment of SBMS at the NMJ.

Similar to βPS, the accumulation of α-Spectrin at synaptic terminals appeared to be limited by neuronal Tnc and promoted by muscle-derived Tnc, as indicated by knockdown analyses (*Figure 6— figure supplement 1A–D*). Moreover, the changes in synaptic α-Spectrin levels followed the variations in integrin levels at synaptic terminals, since manipulations of *mys/(βPS)* elicited a similar profile for the synaptic α-Spectrin (*Figure 6—figure supplement 1E–H*). Together our data suggest that the postsynaptic Tnc/integrin complexes function to recruit the SBMS at synaptic terminals; this activity seems restricted by neuron-derived Tnc, which appears to limit the accumulation and function of postsynaptic complexes. Complete loss of Tnc triggers severe reduction of integrin and α-Spectrin at synaptic terminals, consistent with the observed disruption of bouton architecture.

## Overexpression of Tnc disrupts postsynaptic βPS integrin and spectrin

Since neuron-derived Tnc recruits βPS integrin in the motor neurons to modulate neurotransmitter release (*Figure 3*), we next examined whether muscle Tnc functions similarly in cis to recruit postsynaptic integrin and SBMS and ensure bouton integrity. For this rescue experiment we used a wide range of Tnc levels and monitored both βPS accumulation and postsynaptic function (Ib bouton area) (*Figure 7A–G*). Paneuronal expression of Tnc rescued the postsynaptic βPS accumulation at *tnc* mutant NMJs; however, these animals had small, *tnc*-like boutons. Thus, neuron-derived Tnc can engage postsynaptic βPS to form non-productive complexes that cannot restore the bouton architecture. When low levels of Tnc were provided in the muscle using a weak promoter and low rearing temperatures (*Figure 7D*), βPS accumulation was rescued to levels exceeding the control and bouton size was fully restored. In contrast, high levels of Tnc induced massive lethality and exacerbated the loss of βPS at *tnc* mutant NMJs; these larvae had poorly defined boutons, with almost tubular NMJ branches (*Figure 7E*). Together these results indicate that only low levels of muscle Tnc could rescue the distribution and function of postsynaptic Tnc/integrin complexes, while excess muscle Tnc is toxic. On the other hand, neuron-derived Tnc can recruit and/or stabilize integrin but cannot form fully functional complexes. In support of this interpretation, we found that only low levels of muscle Tnc could fully restore the accumulation of α-Spectrin at *tnc* mutant NMJs (*Figure 7H–M*); neuronal Tnc induced only a modest increase in synaptic α-Spectrin levels.

The fact that the *trans* Tnc/integrin complexes cannot rescue the bouton size suggests that the *cis* and *trans* complexes have different activities. This could be due to different processing or post-translational modifications of Tnc in motor neurons vs. muscles, which have been reported to modulate the activity of ligand-integrin complexes (*Reichardt and Tomaselli, 1991*). Our Western blot analyses did not provide clear evidence for different post-translational modifications for neuron- and muscle-derived Tnc, although the large Tnc-specific band appeared to include multiple species only when Tnc was overexpressed in the muscle (*Figure 7—figure supplement 1*). Alternatively, neuron- and muscle-derived Tnc may be packaged differently and/or associate with molecules that influence their activities. Indeed, Tnc has multiple vWFC domains and RGD-like motifs that could enable a large repertoire of protein interactions.

Our rescue results may also reflect different distributions for neuron- and muscle-derived Tnc: While neuron-secreted Tnc accumulates at synaptic terminals, muscle-secreted Tnc likely distributes throughout the muscle membrane and may sequester integrin away from NMJ locations, further reducing the βPS accumulation at synaptic terminals. To test this possibility, we overexpressed Tnc in an otherwise wild-type genetic background and examined the βPS signals at perisynaptic vs. muscle attachment sites. In these experiments we observed no changes in βPS levels and distribution at muscle attachment sites or costameres (not shown). However, βPS recruitment at larval NMJ was drastically reduced when Tnc was overexpressed in either neurons or muscles (*Figure 8A–E*). Neuronal excess of Tnc produced a reduction of βPS synaptic levels (to 54% from control, p=0.0002, n = 23), and activities, as reflected by the reduction of bouton size compared with the control (transgene only) (*Figure 8F*). Excess Tnc in the muscles practically abolished the perisynaptic βPS signals. With strong muscle drivers (*24B-Gal4* in *Figure 8D*, or *G14-Gal4* in *Figure 8—figure supplement 1*), these larvae showed ribbon-like NMJs with no interbouton/bouton delimitations exceeding the severity of defects observed at *tnc* mutant NMJs (not shown). A significant number of these animals

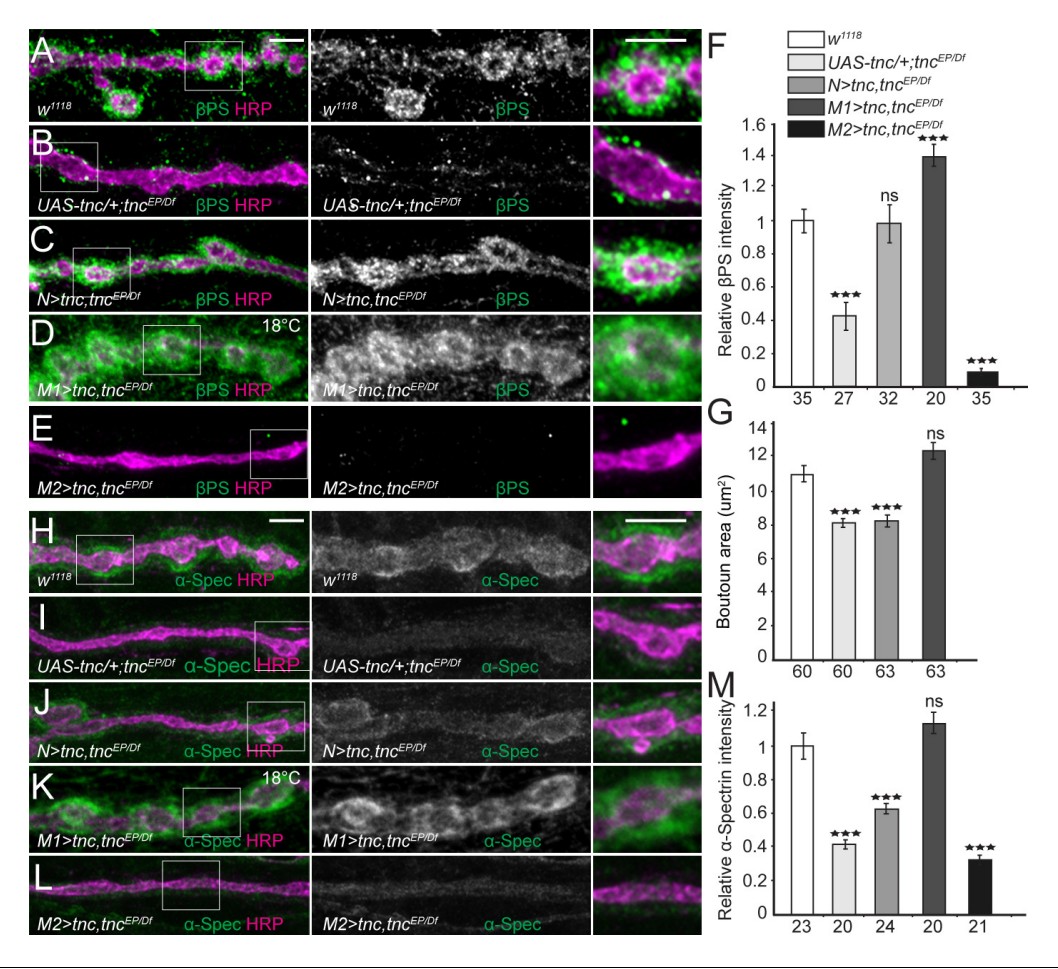

**Figure 7.** Tnc expression restores the synaptic accumulation of βPS at *tnc* mutant NMJs. (**A–M**) Confocal images of NMJ4 boutons of indicated genotypes stained for βPS integrin (**A–E**) or α-Spectrin (**H–L**) (green) and HRP (magenta). The animals were reared at 25°C unless marked otherwise. When expressed in neurons, high levels of Tnc could restore the accumulation of βPS at *tnc* mutant NMJs (quantified in F). However, in these animals the boutons remain small, resembling the *tnc* mutant boutons (quantified in G). For the muscle rescue, Tnc levels were controlled using two different promoters and rearing the animals at 18°C (low expression) or 25°C (moderate). Low levels of muscle Tnc produce substantial accumulation of βPS integrin at *tnc* NMJs, above the control levels, and fully rescued the boutons size; high level of muscle Tnc further decreased the βPS accumulation at *tnc* NMJs. The α-Spectrin synaptic levels are restored only when Tnc is provided at low levels in the muscle. The number of samples examined is indicated in each bar. Bars indicate mean ±SEM. ***p<0.001. Scale bars: 5 μm. Genotypes: control (*UAS-tnc/+;tnc^EP/Df(3R)BSC655*); N > tnc, tnc^EP/Df (*UAS-tnc/+;tnc^EP/elav-Gal4, Df(3R)BSC655*); M1 >tnc, tnc^EP/Df (*BG487-Gal4/UAS-tnc; tnc^EP/Df(3R)BSC655*); M2 >tnc, tnc^EP/Df (*UAS-tnc/+; tnc^EP/24B-Gal4, Df(3R)BSC655*)..
DOI: https://doi.org/10.7554/eLife.35518.017

The following figure supplement is available for figure 7:

**Figure supplement 1.** Western blot analysis of protein lysates from brains or muscles of third instar larvae of indicated genotypes stained for Tnc (green) (predicted MW is 299 kD), and Tubulin (red).
DOI: https://doi.org/10.7554/eLife.35518.018

died during larval and pupal stages and only 46% (n = 211) of third instar larvae developed into adult flies.

The toxicity of excess Tnc during development prompted us to examine the distribution of βPS after a short (8 hr) pulse of Tnc expression in the muscle, using *BG487-Gal4* to induce moderate, gradient muscle expression (*Figure 8G–H'*). This pulse triggered an increase in Tnc immunoreactivies, which distributed diffusely around the synaptic terminals. Importantly the βPS signals were also

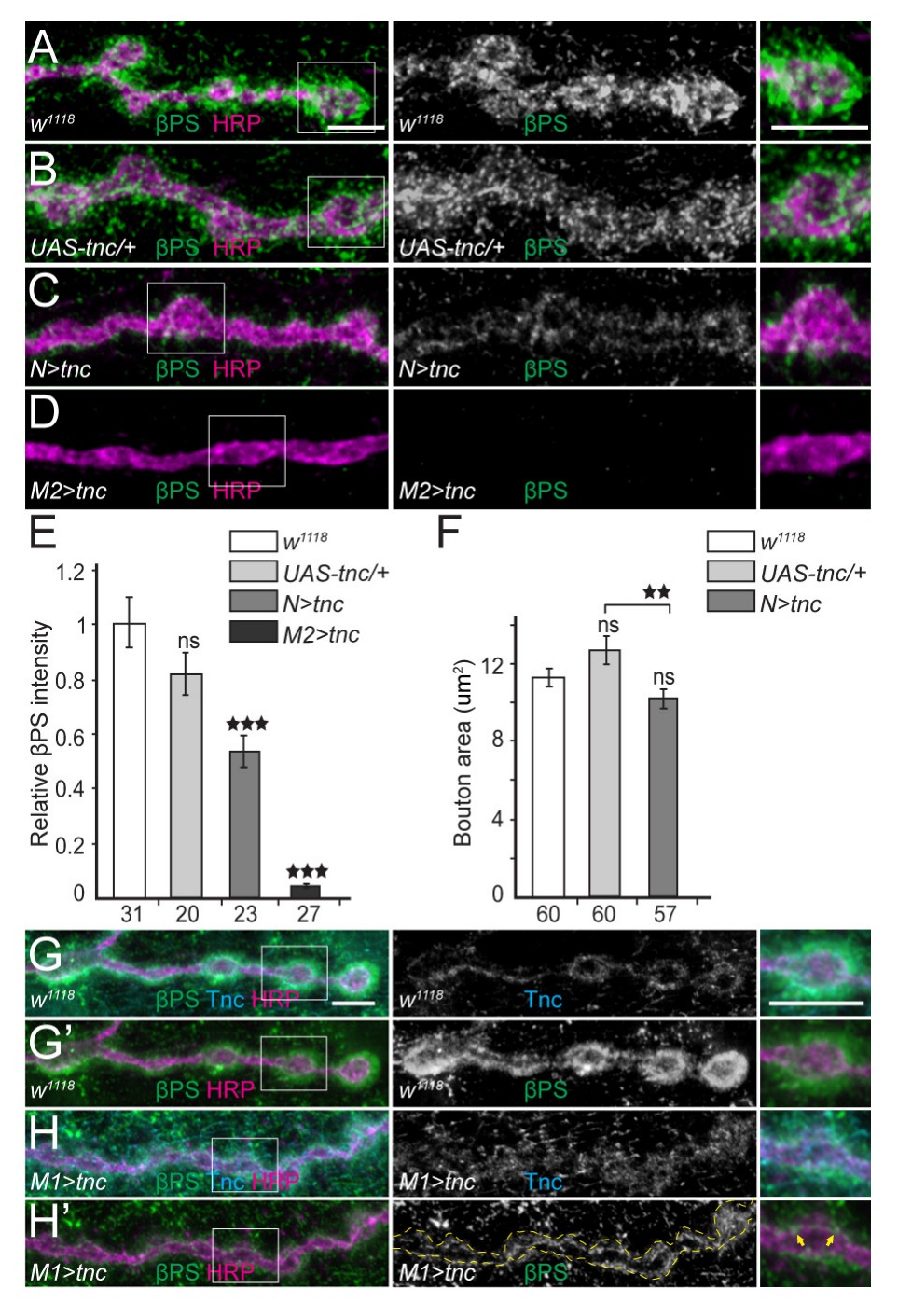

**Figure 8.** Excess Tnc disrupts the βPS accumulation at synaptic terminals. (A–D) Confocal images of NMJ4 boutons of indicated genotypes stained for βPS (green) and HRP (magenta) (quantified in **E**). Excess Tnc induces drastic reductions of synaptic βPS levels compared with controls. Compared with the transgene alone control (**B**), neuronal Tnc significantly reduces the bouton area (quantified in **F**). Overexpression of Tnc in the muscle completely disrupts the NMJ bouton-interbouton boundaries. (**G–H'**) Confocal images of third instar NMJ4 boutons stained for Tnc (Cyan), βPS (green) and HRP (magenta). A pulse of *tnc* expression in larval muscles (in larvae reared at 18°C and shifted at 25°C for 8 hr) increases the levels of Tnc at synaptic terminals and diminishes the postsynaptic βPS signals. The remaining βPS immunoreactivities appear as thin lines (between arrows) localized inside the boutons. The number of samples examined is indicated in each bar. Bars indicate mean ±SEM.
*Figure 8 continued on next page*

*Figure 8 continued*

ns (p>0.05), ***p<0.001, **p<0.01. Scale bars: 5 μm. Genotypes: *N > tnc (UAS-tnc/+;elav-Gal4/+); M1 >tnc (UAS-tnc/BG487-Gal4); M2 >tnc (UAS-tnc/+;24B-Gal4/+).*

DOI: https://doi.org/10.7554/eLife.35518.019

The following figure supplement is available for figure 8:

**Figure supplement 1.** Deleterious effects of excess muscle Tnc.

DOI: https://doi.org/10.7554/eLife.35518.020

drastically diminished, particularly at postsynaptic locations. The βPS signals were no longer concentrated around the synaptic boutons, and instead appeared as thin lines inside the boutons, along the HRP-marked neuronal membrane. Thus, it appears that excess Tnc in the muscle gradually disrupted the postsynaptic βPS accumulation, revealing a small but clear pool of presynaptic βPS; further Tnc overexpression in the muscle completely disrupted both the Tnc and βPS synaptic accumulation and altered the boutons morphology. This dose dependent depletion of Tnc and βPS synaptic signals indicates that excess Tnc may form large aggregates that may be physically excluded from the intercellular space and dispersed away from the synaptic terminals. Alternatively, excess Tnc may trap integrin in the secretory compartment and/or overload a limiting step for the synaptic targeting and recruitment of Tnc/integrin complexes. We favor the former possibility because (i) Tnc itself diffused away from the synaptic terminal when in mild excess, (ii) Tnc has been previously implicated in the formation of large aggregates that fill the lumen of epithelial organs (*Syed et al., 2012*), and (iii) simple disruption of Tnc trafficking and targeting within the muscle cannot explain the loss of presynaptic integrin accumulation and function (*Figure 8D–E* and not shown).

To our knowledge this is the first example where genetic manipulations completely abolished the synaptic accumulation of βPS integrin. These larvae also exhibited drastically reduced levels of synaptic α-Spectrin and Adducin (*Figure 8—figure supplement 1*). Together these results indicate that optimal levels of secreted Tnc are required for proper βPS accumulation at the NMJ, which ensures normal NMJ morphology and function. These experiments also uncovered a novel function for βPS integrin in anchoring α-Spectrin at synaptic locations and coupling the ECM of the synaptic cleft (Tnc) with the spectrin-based membrane skeleton.

## Tnc engages integrin and spectrin at the cell membrane

To examine whether Tnc directed the recruitment of integrin and spectrin complexes at the cell membrane, we took advantage of our ability to produce full-length Tnc and Tnc-HA proteins in S2 and S2R + insect cells. Secreted Tnc-HA was concentrated from S2 conditioned media, affinity coupled to Neutravidin beads of 1 μm diameter, then presented to S2R + cells, which express relatively high levels of integrin and spectrin. Unlike control beads (coated with an unrelated HA-tagged protein), Tnc-HA-coupled beads induced a local accumulation of βPS at the periphery of S2R + cells, in the close proximity of the beads (*Figure 9A–B*). This local recruitment of βPS was not caused by mechanical stress, since control beads, either alone or in clusters, did not trigger βPS accumulation. The βPS recruitment was dose-dependent, as beads with variable Tnc-HA levels elicited proportional βPS accumulation (not shown). The Tnc-coupled beads triggered local recruitment of αPS2 but not αPS1 integrin subunit at the surface of S2R$^+$ cells (*Figure 9C–D*, and not shown). This is consistent with our NMJ observations (*Figure 4*) and with previous reports on small Tnc fragments mediating αPS2/βPS-dependent spreading of S2 cells via RGD and RGD-like motifs (*Fraichard et al., 2010*). Moreover, the Tnc-HA-coupled beads induced similar accumulation of α-Spectrin and Adducin at the bead-cell membrane interfaces (*Figure 9E–J*). Thus Tnc triggers the local recruitment of α-Spectrin and Adducin at the cell membrane. Whether this recruitment is mediated either directly by integrin or indirectly, via other integrin activated scaffolds, remains to be determined. Our attempts to knockdown βPS in S2R + cells, and establish a requirement for integrin for Tnc-induced α-Spectrin recruitment, were hampered by an incomplete RNAi knockdown. Nonetheless, our data establish that Tnc provides a powerful means to recruit and/or stabilize integrin and cortical skeleton components at synaptic terminals.

Although extracellular Tnc binds integrin on cell surfaces, our genetic analyses indicate that in vivo Tnc functions in *cis*. This suggests that Tnc positively affects the surface delivery and/or

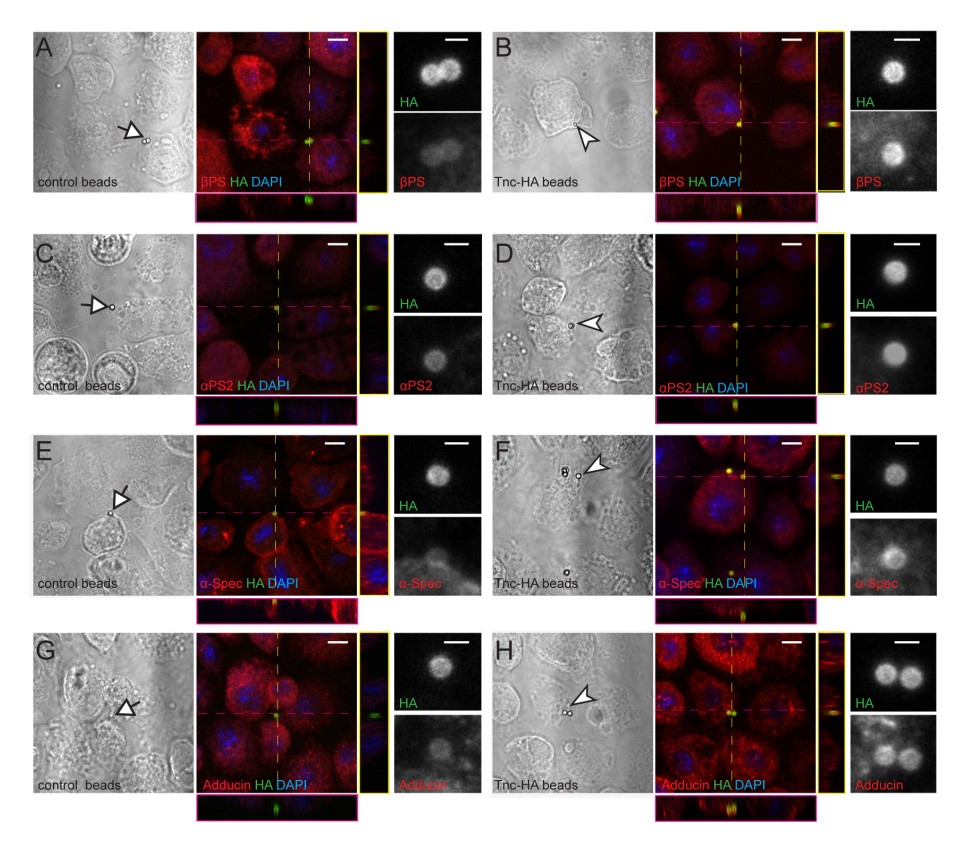

**Figure 9.** Tnc directs accumulation of integrin and spectrin at the cell membrane. (**A–H**) Phase contrast and confocal microscopy images of S2R + cells presented with control (Tld-HA-) and Tnc-HA-coated beads and stained for βPS, αPS2, α-Spectrin or Adducin (in red), HA (green), and DAPI (blue). Arrows indicate control and arrowheads point to Tnc-HA-coated beads, enlarged in details on the right. The Tnc-HA-coupled beads, but not the control, induce strong accumulation of βPS at the bead-cell membrane interfaces, as indicated in the lateral projections, along the dotted lines. The accumulation of αPS2, α-Spectrin or Adducin is also significant in the proximity of Tnc-HA-coupled beads, which appear yellow, in contrast to control beads that remain green. Scale bars: 5 µm, 1 µm in details.

DOI: https://doi.org/10.7554/eLife.35518.021

The following figure supplement is available for figure 9:

**Figure supplement 1.** Cell expressing Tnc have elevated levels of βPS.

DOI: https://doi.org/10.7554/eLife.35518.022

stabilization of βPS integrin. We tested this possibility by following Tnc and βPS levels and distribution in S2R + cells transiently transfected with Tnc (*Figure 9—figure supplement 1*). In this system, Tnc or Tnc-HA were efficiently secreted and accumulated in the media. Inside the cells, Tnc marked large aggregates, likely corresponding to secretory compartments. We found that βPS levels were significantly elevated in cells expressing Tnc; also, βPS co-localized with Tnc in large intracellular aggregates, suggesting that (a) Tnc promotes βPS secretion, and that (b) βPS and Tnc likely associate during intracellular trafficking. It is also possible that excess Tnc may trap βPS inside the secretory compartment, although Tnc appeared to be efficiently secreted and βPS levels were elevated throughout the Tnc-positive cells. Binding to Tnc may stabilize βPS at the cell surface through clustering of integrin complexes and/or by conformational changes and activation of integrins (*Liddington and Ginsberg, 2002*). In both cases, Tnc binding could reduce the rate of βPS integrin endocytosis as well as overall turnover (*López-Ceballos et al., 2016*), which may explain the elevated levels of βPS observed in Tnc-expressing cells.

## Discussion

The ECM proteins and their receptors have been implicated in NMJ development, but their specific roles have been difficult to assess because of their early development functions and the complexity of membrane interactions they engage. In this study, we have shown that Tnc is a selective integrin ligand that enables distinct pre- and post-synaptic integrin activities mediated at least in part through the local engagement of the spectrin-based cortical skeleton. First, Tnc depletion altered NMJ development and function and correlated with selective disruption of αPS2/βPS integrin and spectrin accumulation at synaptic terminals. Second, manipulation of Tnc and integrin in neurons demonstrated that presynaptic Tnc/integrin modulate neurotransmitter release (*Figure 3*); *spectrin* mutations showed similar disruptions of the presynaptic neurotransmitter release (*Featherstone et al., 2001*). Third, postsynaptic Tnc influenced the development of postsynaptic structures (bouton size and SSR complexity), similar to integrin and spectrin (*Figures 3* and *4* and [*Beumer et al., 1999*; *Pielage et al., 2006*]). Fourth, presynaptic Tnc/integrin limited the accumulation and function of postsynaptic Tnc/integrin complexes (*Figures 4*, *5*, *7* and *8*). Fifth, secreted Tnc bound integrin complexes at cell membranes, but only the *cis* complexes were biologically active (*Figures 7* and *9*); *trans* Tnc/integrin complexes can form but cannot function at synaptic terminals and instead exhibited dominant-negative activities (*Figures 7* and *8*). These observations support our model that Tnc is a tightly regulated component of the synaptic ECM that functions in cis to recruit αPS2/βPS integrin and the spectrin-based membrane skeleton at synaptic terminals and together modulate the NMJ development and function.

### Tnc expands the repertoire of ECM synaptic functions

Tnc appears to fulfill unique, complementary functions with the other known synaptic ECM proteins at the *Drosophila* NMJ. Unlike Mtg, which organizes the active zone matrix and the postsynaptic domains (*Rohrbough et al., 2007*; *Rushton et al., 2009*), Tnc does not influence the recruitment of iGluRs and other PSD components. LanA ensures a proper adhesion between the motor neuron terminal and muscle (*Koper et al., 2012*) and also acts retrogradely to suppress the crawling activity-dependent NMJ growth (*Tsai et al., 2012*). The latter function requires the presynaptic βν integrin subunit and phosphorylation of Fak56 via a pathway that appears to be completely independent of Tnc. Several more classes of trans-synaptic adhesion molecules have been implicated in either the formation of normal size synapses, for example Neurexin/Neuroligin, or in bridging the pre- and post-synaptic microtubule-based cytoskeleton, such as Teneurins (*Banovic et al., 2010*; *Mosca et al., 2012*). However, genetic manipulation of Tnc did not perturb synapse assembly or microtubule organization, indicating that Tnc functions independently from these adhesion molecules. Instead, Tnc appears to promote expression and stabilization of αPS2/βPS complexes, which in turn engage the spectrin-based membrane skeleton (SBMS) at synaptic terminals. On the presynaptic side these complexes modulate neurotransmitter release. On the postsynaptic side, the Tnc-mediated integrin and spectrin recruitment modulates bouton morphology. A similar role for integrin and spectrin in maintaining tissue architecture has been reported during oogenesis; egg chambers with follicle cells mutant for either integrin or spectrin produce rounder eggs (*Bateman et al., 2001*; *Ng et al., 2016*).

Our data are consistent with a local function for the Tnc/βPS-recruited SBMS at synaptic terminals; this is distinct from the role of spectrin in endomembrane trafficking and synapse organization (*Kizhatil et al., 2007*; *Lorenzo et al., 2010*; *Tjota et al., 2011*). Embryos mutant for spectrins have reduced neurotransmitter release (*Featherstone et al., 2001*), a phenotype shared by larvae lacking presynaptic Tnc or βPS integrin (*Figures 3* and *4*). However, Tnc perturbations did not induce synapse retraction and axonal transport defects as seen in larvae with paneuronal α- or β- spectrin knockdown (not shown) (*Pielage et al., 2005*). Spectrins interact with ankyrins and form a lattice-like structure lining neuronal membranes in axonal and interbouton regions (*Koch et al., 2008*; *Pielage et al., 2008*; *Goellner and Aberle, 2012*). We found that Tnc manipulations did not affect the distribution of Ankyrin two isoforms (Ank2-L and Ank2-XL) in axons or at the NMJ (not shown); also loss of ankyrins generally induces boutons swelling, whereas Tnc perturbations shrink the boutons and erode bouton-interbouton boundaries. Like *tnc*, loss of spectrins in the striated muscle shows severe defects in SSR structure ([*Pielage et al., 2006*] and *Figure 5*). Lack of spectrins also disrupts synapse assembly and the recruitment of glutamate receptors (*Pielage et al., 2005*;

*Pielage et al., 2006*). In contrast, manipulations of *tnc* had no effect on PSD size and composition (*Figure 5—figure supplement 1*). Instead, *tnc* perturbations in the muscle led to boutons with altered size and individualization and resembled the morphological defects seen in spectrin tetramerization mutants, *spectrin^R22S* (*Khanna et al., 2015*). *spectrin^R22S* mutants have more subtle defects than *tnc,* probably because spectrin is properly recruited at NMJs but fails to crosslink and form a cortical network. Spectrins are also recruited to synaptic locations by Teneurins, a pair of transmembrane molecule that form trans-synaptic bridges and influence NMJ organization and function (*Mosca et al., 2012*). *Drosophila* Ten-m has an RGD motif; we found that βPS levels were decreased by 35% at *ten-m^MB* mutant NMJs (not shown). Thus, Ten-m may also contribute to the recruitment of integrin and SBMS at the NMJ, a function likely obscured by the predominant role both play in cytoskeleton organization.

We have previously reported that α-Spectrin is severely disrupted at NMJs with suboptimal levels of Neto, such as *neto^109*- a hypomorph with 50% lethality (*Kim et al., 2012*). These mutants also had sparse SSR, reduced neurotransmitter release, as well as reduced levels of synaptic βPS ([*Kim et al., 2012*] and not shown). In this genetic background, lowering the dose of *tnc* should further decrease the capacity to accumulate integrin and spectrin at synaptic terminals and enhance the lethality. This may explain the increased synthetic lethality detected in our genetic screen.

## Tnc and NMJ development

In flies or vertebrates, the ECM proteins that comprise the synaptic cleft at the NMJ are not fully present when motor neurons first arrive at target muscles. Shortly thereafter, the neurons, muscles and glia begin to synthesize, secrete and deposit ECM proteins. At the vertebrate NMJ, deposition of the ECM proteins forms a synaptic basal lamina that surrounds each skeletal myofiber and creates a ~ 50 nm synaptic cleft. In flies, basal membrane contacts the motor terminal in late embryos, but is some distance away from the synaptic boutons during larval stages (*Prokop et al., 1998*). Nonetheless, the NMJ must withstand the mechanical tensions produced by muscle contractions. Our data suggest that Tnc is an ideal candidate to perform the space filling, pressure inducing functions required to engage integrin (*Pines et al., 2012*) and establish a dynamic ECM-cell membrane network at synaptic terminals. First, Tnc is a large mucin with extended PTS domains that become highly O-glycosylated, bind water and form gel-like complexes that can extend and induce effects similar to hydrostatic pressure (*Syed et al., 2008*). In fact, Tnc fills the lumen of several epithelial tubes and forms a dense matrix that acts in a dose-dependent manner to drive diameter growth. Second, the RGD and RGD-like motifs of Tnc have been directly implicated in αPS2/βPS-dependent spreading of S2 cells (*Fraichard et al., 2010*). Third, secreted Tnc appears to act close to the source (*Syed et al., 2012*), presumably because of its size and multiple interactions. In addition to the RGD motifs, Tnc also contains five complete and one partial vWFC domains, that mediate protein interactions and oligomerization in several ECM proteins including mucins, collagens, and thrombospondins (*Bork, 1991*). The vWFC domains are also found in growth factor binding proteins and signaling modulators such as Crossveinless-2 and Kielin/Chordin (*Lin et al., 2005*; *Wharton and Serpe, 2013*) suggesting that Tnc could also influence the availability of extracellular signals. Importantly, Tnc expression is hormonally regulated during development by ecdysone (*Fraichard et al., 2010*). Tnc does not influence integrin responsiveness to axon guidance cues during late embryogenesis; unlike integrins, the *tnc* mutant embryos have normal longitudinal axon tracks (*Figure 1—figure supplement 1* and [*Stevens and Jacobs, 2002*]). Instead, Tnc synthesis and secretion coincide with the NMJ expansion and formation of new bouton structures during larval stages. Recent studies have reported several mucin-type O-glycosyltransferases that modulate integrin signaling and intercellular adhesion in neuronal and non-neuronal tissues, including the *Drosophila* NMJ (*Tran and Ten Hagen, 2013*; *Dani et al., 2014*). Tnc is likely a substrate for these enzymes that may further regulate Tnc activities.

## Uncover novel integrin functions via selective ligands

In flies as in vertebrates, integrins play essential roles in almost all aspects of synaptic development. Early in development, integrins have been implicated in axonal outgrowth, pathfinding and growth cone target selection (*Myers et al., 2011*). In adult flies, loss of αPS3 integrin activity is associated with the impairment of short-term olfactory memory (*Grotewiel et al., 1998*). In vertebrates,

integrin mediates structural changes involving actin polymerization and spine enlargement to accommodate new AMPAR during LTP, and 'lock in' these morphological changes conferring longevity for LTP (*McGeachie et al., 2011*). Thus far, integrin functions at synapses have been derived from compound phenotypes elicited by use of integrin mutants, RGD peptides, or enzymes that modify multiple ECM molecules (*Rohrbough et al., 2000*; *McGeachie et al., 2011*; *Dani et al., 2014*). Such studies have been complicated by multiple targets for modifying enzymes and RGD peptides and by the essential functions of integrin in cell adhesion and tissue development.

In contrast, manipulations of Tnc, which affects the selective recruitment of αPS2/βPS integrin at synaptic terminals, have uncovered novel functions for integrin and clarified previous proposals. We demonstrate that βPS integrin is dispensable for the recruitment of iGluRs at synaptic sites and for PSD maintenance. We reveal an unprecedented role for integrin in connecting the ECM of the synaptic cleft with spectrin, in particular to the spectrin-based membrane skeleton. These Tnc/integrin/ spectrin complexes are crucial for the integrity and function of synaptic structures. Our studies uncover the ECM component Tnc as a novel modulator for NMJ development and function; these studies also illustrate how manipulation of a selective integrin ligand could be utilized to reveal novel integrin functions and parse the many roles of integrins at synaptic junctions.

## Materials and methods

### Fly stocks

To generate the $tnc^{82}$ allele, transposons *PBac[RB]e01245* and *PBac[WH]f00323* were mobilized by FRT-induced recombination as previously described (*Parks et al., 2004*; *Thibault et al., 2004*). The *UAS-tnc* lines were generated by insertion of the *tnc* cDNA in pUAST vector and germline transformation (BestGene, Inc.). The following fly lines were obtained from the Bloomington Stock Center: the P-element lines $tnc^{EP}$ (*P[EPgy2]EY03355*) and *ten-m* mutant $Mi[ET1]Ten-m^{MB10734}$ (*Bellen et al., 2004*); the deficiency lines *Df(3R)BSC-318,,−492, −494,* and *−655* (*Parks et al., 2004*); and the TRIP lines $UAS-tnc^{RNAi}$ (*P[TRIP.HMC05051] attP40*), and $UAS-mys^{RNAi}$ (*P[TRIP.JF02819] attP2*), and $UAS-if^{RNAi}$ (*P[TRiP.HMS01872]attP40*). Additional *tnc*, *mys* and *if* RNAi lines showed similar phenotypes, but were not reported here because of mild effects (such as for $UAS-tnc^{RNAi}$ line from Vienna *Drosophila* RNAi Center, *P[GD14952]v42326*), or aberrant muscle development (in the case of *mys* and *if* knockdown). Other fly stocks used in this study were previously described: $neto^{109}$ (*Kim et al., 2012*); *UAS-Dlg* (*Budnik et al., 1996*); *24B-Gal4, elav-Gal4, G14-Gal4, BG487-Gal4.*

Full-length cDNA for *tnc* was assembled in pRM-Tlr plasmid (*Serpe and O'Connor, 2006*) from the following elements: (1) a synthetic PinAI/SalI fragment joining the Tolloid-related signal peptide with the 5' of *tnc* CDS; (2) SalI 7 kb fragment isolated from BACPAC CH322-177A22 (*Venken et al., 2009*); (3) SalI/XhoI PCR product covering the 3' of *tnc* CDS. We used this plasmid to introduce two HA tags in two steps: (1) replace the SalI 7 kb fragment with a pair of primers that introduced 2xHA tags separated by a SalI site; (2) reintroduce SalI kb fragment. The primers used here were:

For-5'-Phos-  TCGAGCTATCCCTATGACGTCCCGGACTATGCACAGTCGACTACCCGTACGATG TGCCCGATTACGCAC and Rev: 5'-Phos-

TCGAGTGCGTAATCGGGCACATCGTACGGGTAGTCGACTGTGCATAGTCCGGGACGTCA TAGGGATAGC. All constructs were verified by DNA sequencing. *UAS-tnc* lines were generated by insertion of the *tnc* cDNA in pUAST and germline transformation (BestGene, Inc.).

To measure the climbing ability, a custom device to fractionate populations based on negative geotaxis was used (*Benzer, 1967*). The flies were placed in the first tube (numbered 0) and left to recover for 2 hr. The fractionation consisted of sequential cycles of tapping down the flies, and moving the flies that climbed above the threshold in 15 s to the next tube. The climbing index was calculated by the formula $(1xN^1 + 2xN^2 + 3xN^3 + 4xN^4 + 5xN^5)/N$, where $N^r$ is the number of flies in fraction r, and N is the total number of flies.

### Protein and tissue culture analyses

*Drosophila* S2 and S2R+ cells were used for expression of recombinant proteins and immunohistochemistry as described previously (*Serpe et al., 2008*). For protein analysis, wandering third instar larvae were dissected, and brains or the body wall (muscle and cuticle) were isolated. The tissues were mechanically disrupted and lysed in lysis buffer (50 mM Tris-HCl, 150 mM NaCl, 1% Triton

X-100, 1% deoxycholate, protease inhibitor cocktail (Roche) for 30 min on ice. The lysates were separated by SDS-PAGE on 4–12% NuPAGE gels (Invitrogen) and transferred onto PVDF membranes (Millipore). The rabbit polyclonal anti-Tnc antibodies were generated as previously described (*Fraichard et al., 2006*) against a synthetic peptide (APVQEYTEIQQYSEGC) (Pacific Immunology Corp.) and affinity purified. Primary antibodies were used at the following dilutions: rabbit anti-Tnc 1:1,000; anti-Tubulin (Sigma-Aldrich), 1:1000; mouse anti-α-Spectrin (3A9), 1:50. Immune complexes were visualized using secondary antibodies coupled with IR-Dye 700 or IR-Dye 800 followed by scanning with the Odyssey infrared imaging system (LI-COR® Biosciences).

Neutravidin beads of 1 μm diameter (Life Technologies) were washed in PBS and preloaded for 30 min at room temperature (RT) with biotin conjugated anti-HA antibody (clone 3F10, Roche). An unrelated HA-tagged protein, Tolloid (Tld), was used as a control (*Serpe and O'Connor, 2006*). Tld-HA and Tnc-HA were concentrated from S2 conditioned media using Amicon Ultra filters (10 kDa) and bound to the preloaded beads overnight. The beads were washed with PBS and presented for 1.5 hr to S2R + cells cultured on chambered coverglass (Thermo Fisher) at 26°C. Cells were next fixed with 4% PFA for 20 min and stained using standard procedures as described below.

## Immunohistochemistry

Wandering third instar larvae were dissected as described previously in ice-cooled $Ca^{2+}$-free HL-3 solution (*Stewart et al., 1994*; *Budnik et al., 2006*). Embryos were collected and fixed using standard procedures (*Patel, 1994*). Primary antibodies from Developmental Studies Hybridoma Bank were used at the following dilutions: mouse anti-GluRIIA (8B4D2), 1:100; mouse anti-Dlg (4F3), 1:1000; mouse anti-Brp (Nc82), 1:200; mouse anti-α-Spectrin (3A9), 1:50; mouse anti-FasII (1D4), 1:10; mouse anti-Futsch (22C10), 1:100; mouse anti-Adducin (1B1), 1:50; mouse anti-βPS integrin (CF.6G11) 1:10; mouse anti-αPS1 integrin (DK.1A4) 1:10; mouse anti-αPS2 integrin (CF.2C7) 1:10. Other primary antibodies were utilized as follow: rabbit anti-Tnc 1:100; rat anti-HA (3F10) (*Lu and Roche, 2012*) 1:500; rabbit anti-FAK (phospho Y397) (Abcam, ab39967), 1:100; rabbit anti-GluRIIB, 1:2000; rabbit anti-GluRIIC, 1:1000 (*Ramos et al., 2015*); rabbit anti-PAK, 1:5000 (a gift from Nicholas Harden) (*Conder et al., 2004*); 1:1000; rat anti-Neto, 1:1000 (*Kim et al., 2012*); and Cy5- conjugated goat anti-HRP, 1:1000 (Jackson ImmunoResearch Laboratories, Inc.). Alexa Fluor (AF) 405-, AF488-, AF568-, and AF647- conjugated secondary antibodies (Molecular Probes) were used at 1:200. All samples were mounted in ProLong Gold (Invitrogen).

Samples of different genotypes were processed simultaneously and imaged under identical confocal settings in the same imaging session with a laser scanning confocal microscope (CarlZeiss LSM780). All images were collected as 0.2 μm optical sections and the z-stacks were analyzed with Imaris software (Bitplane). To quantify fluorescence intensities synaptic ROI areas surrounding anti-HRP immunoreactivities were selected and the signals measured individually at NMJs (muscle 4, segment A3) from ten or more different larvae for each genotype (number of samples is indicated in the graph bar). The signal intensities were calculated relative to HRP volume and subsequently normalized to control. Boutons were counted in preparations double labeled with anti-HRP and anti-Dlg. Bouton size was estimated by using the ImageJ software. All quantifications were performed while blinded to genotype. The numbers of samples analyzed are indicated inside the bars. Statistical analyses were performed using the Student t-test with a two-tailed distribution and a two-sample unequal variance. Error bars in all graphs indicate standard deviation ±SEM. ***p<0.001, **p<0.005, *p<0.05, ns- p>0.05.

## Proximity Ligation Assay (PLA)

PLA was performed following published protocols (*Wang et al., 2015*). In brief, wandering third instar larvae were dissected, fixed and incubated with primary antibodies (anti-Tnc and anti-βPS integrin) overnight at 4°C. To detect the markers, the samples were incubated with AF-conjugated secondary antibodies (anti-rabbit- AF488, anti-mouse-AF405 and anti-HRP-AF647) for 1 hr in the dark. For PLA, the samples were washed with Wash Buffer A (DUO 92101 Kit, Sigma), incubated first with PLA probe anti-mouse MINUS and PLA probe anti-rabbit PLUS (1:5 dilution) for 2 hr at 37°C, then with 200 μl Ligation solution for 1 hr at 37°C, and finally with Amplification solution for 2 hr at 37°C. After washes in Wash Buffer B, the samples were mounted in ProLong Gold (Invitrogen) and examined by confocal imaging.

## Electrophysiology

The standard larval body wall muscle preparation first developed by *Jan and Jan (1976)* (*Jan and Jan, 1976*) was used for electrophysiological recordings (*Zhang and Stewart, 2010*). Wandering third instar larvae were dissected in physiological saline HL-3 saline (*Stewart et al., 1994*), washed, and immersed in HL-3 containing 0.5 or 0.8 mM $Ca^{2+}$ using a custom microscope stage system (*Ide, 2013*). The nerve roots were cut near the exiting site of the ventral nerve cord so that the motor nerve could be picked up by a suction electrode. Intracellular recordings were made from muscle 6, abdominal segment 3 and 4. Data were used when the input resistance of the muscle was >5 MΩ and the resting membrane potential was between −60 mV and −70 mV. The input resistance of the recording microelectrode (backfilled with 3 M KCl) ranged from 20 to 25 MΩ. Muscle synaptic potentials were recorded using Multiclamp 700B amplifiers (Molecular Devices) and Axon Clamp 2B amplifier (Axon Instruments) and analyzed using pClamp 10 software. Following motor nerve stimulation with a suction electrode (200 μsec, 1.9 V), evoked EJPs were recorded. Four to six EJPs evoked by low frequency of stimulation (0.1 Hz) were averaged. For mini recordings, TTX (1 μM) was added to prevent evoked release (*Stewart et al., 1994*). To calculate mEJP mean amplitudes, 100 events from each 10 or more NMJs (only one NMJ per animal was used) were measured and averaged using the Mini Analysis program (Synaptosoft). Minis with a slow rise and falling time arising from neighboring electrically coupled muscle cells were excluded from analysis (*Gho, 1994*; *Zhang et al., 1998*). Paired stimuli (200 μsec, 1.9 V, 0.05 Hz) were applied with a suction electrode at 50 ms inter-stimulus intervals. The amplitude of the eEJP was determined as an average from 5 to 8 steady consecutive sweeps. The paired-pulse ratio (PPR) was expressed as the amplitude ratio of the second synaptic response to the first synaptic response (*Zhang and Stewart, 2010*). Quantal content was calculated by dividing the mean EJP by the mean mEJP after correction of EJP amplitude for nonlinear summation according to previously described methods (*Stevens, 1976*; *Feeney et al., 1998*). Corrected EJP amplitude = E[Ln[E/(E - recorded EJP)]], where E is the difference between reversal potential and resting potential. The reversal potential used in this correction was 0 mV (*Feeney et al., 1998*; *Lagow et al., 2007*). Statistical analysis was performed with Kaleida-Graph 4.5 (Synergy Software) using ANOVA followed by a Tukey post hoc test. Differences were considered significant at $p < 0.05$. Data are presented as mean ±SEM.

## Electron microscopy

Wandering third instar larvae were dissected in physiological saline HL-3 saline and fixed for 30 min on dissection plate in fixation buffer (0.1 M Sodium Cacodylate buffer, pH7.2; 2 mM $MgCl_2$; 1% glutaraldehyde; 4% paraformaldehyde). The samples were trimmed to include only the abdominal segments A2 and A3, transferred in a 1.5 mL Eppendorf tube, fixed overnight at 4 °C, then washed extensively with 0.1 M Sodium Cacodylate buffer with 132 mM Sucrose, pH 7.2. The samples were further processed and analyzed according to published protocols (*Ramachandran and Budnik, 2010*) at the Microscopy and Imaging Facility, NICHD.

## Acknowledgements

This work was supported by the Intramural Research Program at the NICHD, NIH. We thank Bing Zhang, Graham Thomas, Herman Aberle, Kelly Ten Hagen, Ed Giniger, and Alan Hinnebusch for helpful discussions and suggestions. We are grateful to Nicholas Harden for antibodies. We thank Tom Brody, Rosario Vicidomini and Lindsey Friend for comments on this manuscript. Electron microscopy was in part performed at the Microscopy and Imaging Core of NICHD, NIH with the assistance of Chip Dye. We also thank the Bloomington Stock Center at the University of Indiana for fly stocks and the Developmental Studies Hybridoma Bank at the University of Iowa for antibodies.

## Additional information

### Funding

| Funder | Grant reference number | Author |
|---|---|---|
| NIH Office of the Director | Z01 HD008914 | Qi Wang<br>Tae Hee Han<br>Peter Nguyen<br>Mihaela Serpe |
| NIH Office of the Director | Z01 HD008869 | Qi Wang<br>Tae Hee Han<br>Peter Nguyen<br>Mihaela Serpe |

The funders had no role in study design, data collection and interpretation, or the decision to submit the work for publication.

### Author contributions

Qi Wang, Conceptualization, Formal analysis, Investigation, Visualization, Methodology, Writing—original draft, Writing—review and editing; Tae Hee Han, Formal analysis, Investigation, Visualization, Writing—original draft; Peter Nguyen, Data curation, Validation, Investigation, Methodology; Michal Jarnik, Investigation, Visualization; Mihaela Serpe, Conceptualization, Supervision, Investigation, Writing—original draft, Project administration, Writing—review and editing

### Author ORCIDs

Mihaela Serpe ⓘ http://orcid.org/0000-0002-9205-8589

### Decision letter and Author response

Decision letter https://doi.org/10.7554/eLife.35518.025
Author response https://doi.org/10.7554/eLife.35518.026

## Additional files

### Supplementary files

• Transparent reporting form
DOI: https://doi.org/10.7554/eLife.35518.023

### Data availability

All data generated or analysed during this study are included in the manuscript and supporting files.

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
