## [Decision Letter]

[Editors’ note: a previous version of this study was rejected after peer review, but the authors submitted for reconsideration. The first decision letter after peer review is shown below.]

Thank you for submitting your work entitled "Tenectin recruits integrin to stabilize boutons and regulate vesicle release at the *Drosophila* neuromuscular junction" for consideration by *eLife*. Your article has been reviewed by three peer reviewers, and the evaluation has been overseen by a Reviewing Editor and a Senior Editor. The following individuals involved in review of your submission have agreed to reveal their identity: Andreas Prokop (Reviewer #2).

Our decision has been reached after consultation between the reviewers. Based on these discussions and the individual reviews below, we regret to inform you that your work will not be considered further for publication in *eLife*.

All three reviewers (see below their comments) expressed major criticism regarding the manuscript conclusions. The major comment is that the results are confusing and some time contradicting regarding the muscle, versus neuron-dependent function of Tenectin in recruiting integrins to the NMJ. In addition a direct versus indirect effect of Tenectin on the levels of Spectrins at the NMJ should be further validated. Many controls are also missing.

We hope you find our comments useful. If you address these comments and would like to make a fresh submission at *eLife*, rather than elsewhere, we will be happy to examine this afresh.

Reviewer #1:

This paper describes the expression and function of Tenectin at the larval NMJ. While there is a lot of information in the paper, it will primarily be of interest to specialists in fly neuromuscular system development.

There are also some problematical aspects to the data, as follows:

1) Maybe I am not understanding something here, but there seems to be an unresolved contradiction about *cis* vs. *trans* action of Tnc. They show that muscle, but not neuronal Tnc knockdown reduces βPS expression (Figure 4M) and from this (subsection “Muscle Tnc recruits postsynaptic integrin”, second paragraph) they conclude that it muscle Tnc acts in *cis* to recruit βPS, which is primarily expressed in muscles.

However, in Figure 6 they show that in a *tnc* mutant the loss of expression of βPS is restored by neuronal expression of Tnc, indicating that it is acting in *trans* (Figure 6D). When they express Tnc in muscles they lose βPS expression. The results in Figure 4 would predict that neuronal expression of Tnc in the mutant should not rescue, since neuronal knockdown does not affect βPS, and that muscle expression of Tnc should rescue, since muscle knockdown decreases βPS. Perhaps this is a Tnc overexpression phenotype. To test this, they should use weaker muscle drivers and reduce the temperature to reduce GAL4 activity, or make a direct muscle promoter>Tnc fusion. Unless I am missing something here, if they can express Tnc at physiological levels in the muscles in a *tnc* mutant, this should rescue βPS expression, not eliminate it.

2) The data in Figure 8 are not convincing. Looking at the high-mag views of the HA beads and placing them into the low-mag views of the cells, one can see that there is no accumulation of βPS, αPS2, or spectrin around the beads relative to their levels elsewhere in the cells. In fact the beads are in low-expression areas for all of these markers. Given this, one cannot believe that the levels of these markers are lower around the control beads. No high-mag views of the control beads are shown, and you can't even see these beads in the fluorescent images since they are not labeled. To do this experiment correctly, they need to use a different HA-tagged protein on the control beads, so that they can be seen in the fluorescent images, and show equivalent high-mag views of the beads so that it can be determined if there is really more integrin around the beads. This should also be quantitated by measuring fluorescence intensity in the bead regions in Tnc-HA vs. Control protein-HA beads.

3) Figure 8—figure supplement 1 is also unconvincing and incomplete. There is clearly a lot of βPS within the cell in Tnc-HA expressing cells, but levels at the cell surface are not clearly increased (at least the images shown do not demonstrate this). They could address this by staining live cells with anti-βPS without detergent. Maybe Tnc-HA is expressed at very high levels and sequesters βPS within a secretory compartment where it cannot be degraded, thus increasing its intracellular levels. This does not show that Tnc-HA acts in *cis* to increase integrin surface localization.

4) If they want to evaluate whether Tnc-HA induces βPS, they should make a stable S2 line expressing Tnc-HA, which is easily done using stable transformation vectors. If Tnc-HA expression is toxic they can drive it from a conditional promoter. Similarly, they can use stable transfection to knock down βPS with RNAi if the transient RNAi knockdown is inadequate (subsection “Tnc engages integrin and spectrin at the cell membrane”, first paragraph).

Reviewer #2:

This work by Wang and colleagues has some clear merits and high potential, but I do not recommend publication, but rather rejection with the option to re-submit. The main reason for this suggestion is the lack of clarity across the paper how experiments link together into new understanding. In my view there are seemingly contradicting results in this work that suggest a very intricate role and regulation of *tnc*, but have to be resolved before this manuscript can be considered for publication. The contradictions mainly consist in obscure effects observed upon pre- versus post-synaptic manipulations that require a far more refined experimental approach and logic dissection. Instead, data are being uncritically reported and the reader is left alone with a highly confusing message, if any message at all. There is potential in many of the data, but the authors failed to turn them into a readable and informative manuscript. Apart from these shortcomings there are a number of further issues that need clarification. See my detailed comments below.

Detailed comments:

The Introduction reads well and provides a sound overview of ECM at mammalian and *Drosophila* synapses. It would be helpful to mention the obvious gaps of understanding in the field and how this work addresses these gaps. A tip: do the authors know this review: Singhal, N., Martin, P. T. (2011). Role of extracellular matrix proteins and their receptors in the development of the vertebrate neuromuscular junction. Dev Neurobiol 71, 982-1005 – https://doi.org/10.1002/dneu.20953

Introduction or start of Results: It needs to be clearly stated that *tnc* has no mammalian homologue.

“Using a polyclonal anti-Tnc antibody (Materials and methods) we found that Tnc signals are concentrated during late embryogenesis in the ventral cord, near the pioneer, anti-Fasciclin II (FasII) positive axons (Figure 1—figure supplement 1A, B and (Fraichard et al., 2006)): For this statement, the resolution of analyses is far too low, and I do not think that this statement is important. Delete and stay with the statement that staining it is strongly enriched in the neuropile.

“In contrast, the Tnc CNS signals were absent in a *tnc* mutant (*tnc^EP^* – *P[EPgy2]EY03355*), predicted to disrupt both known *tnc* transcripts (Syed et al., 2012)”: Point out that Fas2 pattern is not disturbed in the mutant CNS. Obviously, there is not interaction with reported functions of integrins in this context:

Stevens and Jacobs, 2002; Broadie, Baumgartner and Prokop, 2011. *Drosophila*

Subsection “Tnc localizes at the synaptic cleft”, second paragraph: indicate the two *tnc* transcripts in Figure 1A.

Subsection “Tnc localizes at the synaptic cleft”, second paragraph and thereafter: the correct term is "hetero-allelic": if *tncDf* is a true deficiency you need to talk of a "hemizygous" condition.

Subsection “Tnc localizes at the synaptic cleft”, third paragraph”: At this early point, the pre-postsynaptic statement cannot be made. Tnc expression in the neuropile is in patches which might reflect staining around dendrites. Therefore, the expression/localisation in motorneuons may well be postsynaptic (equivalent to muscle expression in the periphery).

Subsection “Tnc localizes at the synaptic cleft”, third paragraph”: Does complete knock-down in the CNS get rid of all expression or is *tnc* only expressed in motorneurons but not very efficiently knocked down? Along these lines, given potentially expected different driver strengths in CNS and muscles, the conclusions about where *tnc* is strongest expressed cannot be drawn from the knock-down data.

Subsection “Tnc localizes at the synaptic cleft”, last paragraph”: the staining does not look like being around nuclei, but seems suprisingly nuclear. If it were in Golgi or ER, it would have a broader expression.

Subsection “Tnc localizes at the synaptic cleft”, last paragraph”: the reduced staining is not very convincing. This needs to be quantified.

Subsection “Tnc localizes at the synaptic cleft”, last paragraph: either show and quantify the *tnc* over-expression data or take them out.

Figure 1—figure supplement 2 is not well described in its legend and is confusing. What nuclei are shown on the right? I guess it is muscle nuclei? But why is there no expression in those when driving *tnc* with 24B (bottom right image). Please, clarify, because at the moment this does not make sense. Expression data *must* be quantified across these experiments.

Subsection “*tnc* mutants have impaired NMJ physiology”, first paragraph: either show larval turning assay results or take the statement out.

Why were white flies chosen for climbing assays? At what age of flies were these experiments performed? An independent experiment with knock-down of *tnc* would be helpful.

Figure 2: indicate the statistical test used. Indicate in your sample number also how many events were included in the statistics: e.g. 200/13 (200 events from 13 NMJs).

Subsection “*tnc* mutants have impaired NMJ physiology”, last paragraph: mention that reduced mEJC frequency is also in agreement with reduced quantal content due to less release sites.

Subsection “*tnc* mutants have impaired NMJ physiology”, last paragraph: "as expected", provide references for this statement.

Subsection “*tnc* mutants have impaired NMJ physiology”, last paragraph: what interval length was used? This is not indicated in text or figure. Ideally, a plot with increasing length should be shown to illustrate that this is true short-term depression. In the graph indicate that you show the ratio of 1st/2nd pulse

Subsection “*tnc* mutants have impaired NMJ physiology”, last paragraph: It needs a clear explanation as to why short-term facilitation explains decreased probability. I can come up with my own speculations, but it is the task of the authors to make this clear.

“Both pre- and postsynaptic components modulate neurotransmitter release”: where does this statement come from? It is confusing without further explanation, especially since it is followed by posing this problem and since it opposes the title. Should this be the title of the section?

Subsection “Neuronal Tnc modulates neurotransmitter release”, first paragraph: why "interestingly"? Is this not expected if presynaptic expression leads to release at NMJs and mutants show reduced frequency? Please, build a logic and solid argumentation.

In all graphs, indicate absence of tested significance with a symbol, such as "ns". In this way it is clear which bars were assessed that the asterisks have not simply been forgotten to insert.

Subsection “Neuronal Tnc modulates neurotransmitter release”, last paragraph: the term "recruits" is not supported by the data; at this point you can merely state that *tnc* seems to act as a ligand for presynaptic integrin.

Figure 3G. I wonder whether only one set of experiments is sufficient here. Reproducing the data through knock-down of talin or inflated would clearly strengthen these observations.

Subsection “Muscle Tnc recruits postsynaptic integrin”, first paragraph: the co-localisation is not convincing and the authors *must* refine their argumentation. There is too much expression of integrin to make such a statement. Any correlation analyses of *tnc* would likely show overlap, but there is much expression of integrin that does not correlate. Even more, these data (double-labelling with HRP) clearly confirm previous observations that integrin localises also strongly to SSR, i.e. postsynaptic membrane. Most of the apparent overlap occurs therefore in areas that are irrelevant for the presynaptic function, as demonstrated by lack of effect upon postsynaptic integrin or *tnc* knock-down.

Figure 4: to address the presynaptic localisation of integrins, it would be very helpful to knock down integrins postsynaptically and see that staining remains in a sharp line only around boutons – which should then vanish in *tnc* mutant background. Please, combine G and H into one graph (both are normalised anyway)

Subsection “Muscle Tnc recruits postsynaptic integrin”, second paragraph: "secreted into the synaptic cleft"?

Subsection “Muscle Tnc recruits postsynaptic integrin”, second paragraph: loss of βPS upon muscle but not neuronal knock-down of *tnc* is in stark contrast to data in Figure 1—figure supplement 2 where only neuronal *tnc* sheds into the synaptic cleft. The stabilisation theory based on ligand binding does therefore not work.

Subsection “Muscle Tnc recruits postsynaptic integrin”, second paragraph: How can the authors suddenly favour postsynaptically derived Tnc, although all their data from before show a physiological relevance only for neuronal Tnc? I am completely lost at this point.

Subsection “Muscle Tnc recruits postsynaptic integrin”, last paragraph: "is required for the recruitment of PAK" – the statement as is, suggests a direct mechanism.

Subsection “Muscle Tnc recruits postsynaptic integrin”: the last sentence is too condensed.

Figure 5: I do not believe the EM data. How many muscles from how many independent animals were analysed? Were larvae processed in the same vial? Image quality is very poor and there is a clear osmolarity effect; from anti-FAK staining one gets the impression that SSR is smaller, yet here it is blown up in mutants. Analyses have to be performed in whole mounts with SSR markers (for example anti-DLG) and EM analyses have to be made more transparent to the reader and properly quantified. The odd shape of boutons looks not more than the section being almost tangential to the bouton. Shape statements are far easier made from whole mount stainings.

Subsection “*tnc* mutants have disrupted spectrin-based membrane skeleton”, third paragraph: there are no analyses of bouton sizes in the paper, which *must* be provided for mutant and upon pre- and post-synaptic knockdown of *tnc*.

Subsection “*tnc* mutants have disrupted spectrin-based membrane skeleton”: to my knowledge, loss of spectrin causes a cell-autonomous defect in neuronal terminals, i.e. the change in boutons is not caused by loss of spectrin in muscles. This does not fit with the argumentation of the authors.

Subsection “Overexpression of Tnc disrupts postsynaptic βPS integrin and spectrin”: the manuscript gets utterly confusing now, and the authors make no effort to shed light into this. It becomes close to unreadable. There seems to be a constant contradiction between pre- and postsynaptic requirements which are not resolved.

Subsection “Overexpression of Tnc disrupts postsynaptic βPS integrin and spectrin”, second paragraph: please, explain what the point of these experiments is? What do we learn? This is an artificial situation that does not tell us much.

Discussion, first paragraph: integrins are very well known to be locally activated through ligands, so I do not understand this claim.

Reviewer #3:

The manuscript "Tenectin recruits integrin to stabilize bouton architecture and regulate vesicle release at the *Drosophila* neuromuscular junction" represents an interesting investigation of the role of a secreted EMC protein at the NMJ. The manuscript provides novel information on the role of Tnc in recruiting the integrin complex (αPS2/βPS) to the NMJ along with spectrin and adducin. Overall the manuscript presents interesting and novel insights into the role of Tnc and integrin complexes at the NMJ. Much of the data and the quantitation was convincing and supported the conclusions. However there are a number of points that need to be addressed with the explanation of the data, some of the major conclusions drawn, a need for discussion of alternative hypotheses and the confirmation of the protein complex in vivo. These points are outlined below:

For the RNAi experiments only one *tnc^RNAi^* line and one *mys^RNAi^* was used throughout. Neither of these lines have been previously published to be specific to Tnc or *mys* and thus this either needs to be proven or alternative previously verified lines used to ensure that there are not any off target effects. Specifically the experiments need at least two RNAi lines used for the Tnc and *mys* knockdown to ensure no off target effects or RNAi lines verified by others to be specific. The *mys^RNAi^* for instance is presumably a weaker one given that muscle knockdown of *mys* if strong would result in embryonic lethality.

In Figure 3 comparing mEJP Frequency for the *tnc^EP/Df^*was less strongly affected compared to the *N>tnc^RNAi^* in Figure 3E versus 3B. It is surprising that the RNAi would have a strong effect that the EP/Df combination especially given that in Figure 1 the protein levels were substantively reduced in the EP/Df but not the *N>tnc^RNAi^*. This disconnect between levels of protein versus phenotypes need to be addressed in the text.

With the residual Tnc immunolabeling present in the *tnc^EP/Df^*, it is logical to conclude that there is significantly more Tnc present in the *N>tnc^RNAi^* NMJ. However the degree of reduction of Tnc at the NMJ is not shown with either *N>tnc^RNAi^* or *M>tnc^RNAi^*. This should be included or if not, explained. This becomes very relevant in Figure 4 as the effect of *M>tnc^RNAi^* on the recruitment of αPS2βPS to the NMJ had a stronger effect than *N>tnc^RNAi^*.

The degree of Tnc loss in the *tnc^EP/Df^* combination in Figure 1G (and Figure 1—figure supplement 1D) was surprisingly not complete given that the allele seems to be a null. The immunolabeling in the image does not match the quantitation by Western seen in Figure 1C. The phrase "much reduced" in *tnc^EP/Df^* could be quantified using the approaches utilized throughout the manuscript for all the other markers. The text should also address if this is truly the presence of residual Tnc at the NMJ (and the implications of this) or is this background?

The results from this manuscript suggest that αPS2βPS and Tnc may directly interact. The S2 cell experiment was a strong approach to understanding the associations of Tnc, the integrin complex, spectrin and adducin. A stronger and a necessary approach to investigating the association of these proteins was to prove that these occur in vivo using a proximity ligation assay (PLA) to determine where in the NMJ these associations happen. The advantage of this approach is that the *tnc^EP/Df^* mutant would serve as a good control as would the different RNAi and overexpression approaches. The PLA experiment would also be able to address if the *M>tnc* expression was able to recruit the αPS2βPS complex to ectopic sites away from the NMJ. This would go a long way to support many of the conclusions and how a clear link at the NMJ between these protein complexes.

Further to this point, the link between integrins to spectrin and adducin is the key point of interest but other than the S2 cells there is little in the way to support a physical link between these components. It is equally possible that the loss of spectrin or adducin itself may lead to the loss of αPS2βPS from the NMJ. What happens to spectrin or adducin when αPS2 is reduced and vice versa?

For Figure 5 it is mentioned that "In the course of these experiments we noted that *tnc* mutant NMJs showed aberrant morphology with poorly defined bouton/interbouton boundaries and more tubular branches, particularly in the proximal region." Given that this is the first mention of these different phenotypes, an inclusion of some examples is warranted as prior indications from Figure 2—figure supplement 1 was there were little morphological changes.

For the TEM analysis comments such as "However, the mutant boutons were drastically distorted and no longer maintained normal round/oval shapes" and "This phenotype was highly penetrant and affected all type 1B neurons" should be supported by the number of boutons that were analyzed and the number of synapses/larvae analyzed in the text itself.

The distribution of Dlg should be shown (data not shown throughout the manuscript) and the quantitation provided, especially given the effects on adducin on Dlg at the NMJ. This later point should be addressed in the text as well.

The *M>tnc^RNAi^* data with α-spectrin showed no decrease compared to the *UAS-tnc^RNAi^* control. While the *N>tnc^RNAi^*showed an upregulation. Yet the *tnc^EP/Df^* displayed a reduction in α-spectrin. The manuscript needs to address these differences and provide a model to help guide through the different interpretations of these effects.

The manuscript makes the statement "Since neuron-derived Tnc recruits βPS integrin in the motor neurons to modulate neurotransmitter release (Figure 3…)" – this is a very strong statement given that *N>tnc^RNAi^*showed a mild increase of the distribution of βPS to the NMJ (Figure 4L, 4N) and the effect on αPS2 is not shown. Figure 3 provides no evidence that the changes observed with *N>tnc^RNAi^* are due to the loss of βPS. This would require a rescue experiment to provide a direct link.

The interpretations and conclusions for the role of neuronal or muscle driven Tnc need to be tempered. *N>tnc* rescued the degree of βPS present at the NMJ in the *tnc^EP/Df^* mutant while *M>tnc* did not and this lead to the conclusion that neuronal Tnc is key to the recruitment of βPS. Yet *N>tnc* alone decreases the βPS at the NMJ (Figure 7C, 7E) compared to controls while *N>tnc^RNAi^* knock down (Figure 4L, 4N) shows slight more βPS at the NMJ compared to control. This suggests that neuronal derived Tnc blocks integrin accumulation at the NMJ. On the muscle side of the equation *M>tnc* cannot rescue the *tnc^EP/Df^* which isn't too surprising given that *M>tnc* removes all the tnc from the NMJ. Thus these experiment don't prove that neuronal derived Tnc is sufficient and necessary to recruit βPS. Especially given that Tnc knockdown in muscles but not in neurons reduced the amount of synaptic βPS.

Along these lines the *tnc* mutants with muscle overexpressed *tnc* had a dramatic reduction in βPS as well as the *M>tnc* in a wild type background. The panels from Figure 1J, Figure 1—figure supplement 1F and Figure 1—figure supplement 2F would suggest that overexpression of *tnc* may be significantly deleterious to the entire NMJ not just the distribution of βPS.

The authors state that expression of *N>tnc* in the *tnc^EP/Df^* mutant was able to rescue the reduction in α-spectrin however the degree of increase in α-spectrin does not appear to be strong. For the quantitation (Figure 6J) it was not clear what was being compared –.…each experimental to control which would suggest that the *N>tnc* was significantly different from control (and didn't rescue to a great extent) –.…or each experimental to each other which would suggest that all three experimental were significant different from each other which didn't appear to be the case. For this type of multiple experiment analysis it would also be more relevant to carry out a One Way Anova with a post hoc multi-comparison test rather than a students' t-test.

It was also intriguing that the degree of rescue of Adducin and βPS immunolabeling at the NMJ was so much better than that for α-spectrin with the *N>tnc*. Given the strength of the statements made in the text is it possible the wrong data was included in Figure 6J?

The major conclusion from this section should be tempered to reflect that neuronal expression of Tnc can rescue the *tnc^EP/Df^* spectrin, βPS and adducin levels. Whether *M>Tnc* is able or not will require a more measured approach to ensure that muscle expressed Tnc is present at the NMJ, that the muscles themselves are not deleteriously affected. This might be case given the stronger effects on βPS by the increased expression of Tnc in wild type muscles compared to the *tnc^EP/Df^* muscles. These effects may simply by an increase the deleterious effects of Tnc expression and thus lead to a great disruption of NMJ morphology rather than an specific effect on "ligand redundancy and/or other mechanisms may allow for partial βPS and/or α-Spectrin synaptic accumulation". Would the increased expression of Tnc or any ECM component lead to disruption of muscles as a result from ER stress?

The statement "Unlike βPS, which was completely lost at these NMJs" is not supported by the images presented in Figure 4E, Figure 6B and 6E: βPS is not completely lost at these NMJs. The authors need to be more careful about making this type of absolute statement.

[Editors’ note: what now follows is the decision letter after the authors submitted for further consideration.]

Thank you for resubmitting your work entitled "Tenectin recruits integrin to stabilize bouton architecture and regulate vesicle release at the *Drosophila* NMJ" for further consideration at *eLife*. Your revised article has been favorably evaluated by K VijayRaghavan (Senior Editor), a Reviewing Editor, and three reviewers.

The manuscript has been improved and the reviewers were impressed by the quality and amount of data. However, there are some remaining issues that need to be addressed before acceptance, as outlined below:

Two of the reviewers requested to add controls, including additional RNAi lines for tenectin, and for the GAL4 used. This can be done speedily. Also, the genetic interaction between βPS and tenectin mutants would further support the results indicated by reviewer 1. Please see if you can do this.

Below, please find the specific comments of three reviewers.

Reviewer #1:

Wang et al. report an interesting finding on distinct pre-synaptic and post-synaptic roles for tenectin (Tnc), a selective integrin ligand, at the *Drosophila* neuromuscular junction (NMJ). Tnc in the neuron interacts with βPS/αPS2 integrin to control neurotransmitter release, whereas Tnc in the muscle interacts with post-synaptic αPS2/βPS integrin to regulate bouton morphology. By manipulating Tnc, they uncovered a novel role for integrin in recruiting the spectrin based membrane skeleton at the NMJ.

This paper will be of broad interest to the readership of *eLife* because it advances our understanding of NMJ development and synaptic physiology, as well as integrin/extracellular matrix biology. Although Tnc has been studied at the *Drosophila* NMJ, Wang et al. is among the first to reveal that Tnc has differential function in neurons and muscle. The data presented in this paper is thorough, with good controls and overall high-quality figures. However, I have two major concerns that should be addressed before publication. First, the inclusion of only one RNAi line targeting Tnc is insufficient, especially given the fact that the neuronal RNAi line only knocks down Tnc levels to 43% of control. The inclusion of at least two RNAi lines showing the same phenotype would both confirm and strengthen the author's conclusions. This is not a detail and needs to be addressed.

Second, the conclusion that αPS2, βPS, and Tnc function together to modulate neurotransmitter release (Figure 2) should be strengthened by genetic interaction experiments between βPS and Tnc indicating that either a) double mutants do not have a more severe phenotype, or b) trans-heterozygotes show enhancement of phenotype.

Reviewer #2:

Tenectin recruits integrin to stabilize bouton architecture and regulate vesicle release at the *Drosophila* NMJ' Wang et al.

In this manuscript the authors identify the mucin class protein tenectin as an ECM component of the *Drosophila* NMJ synapse from an interaction screen with Neto. The description of tenectin mutants is comprehensive, the technical quality of the data is high and the authors do identify interesting interactions with integrins. However ultimately little new insight is gained into the role of the ECM in synapse development or function beyond the identification of a new component. As such, the manuscript seems better suited to a more specialised neuroscience journal.

Reviewer #3:

The manuscript by Wang et al. reports a novel function of tenectin (*tnc*) at the neuromuscular junction (NMJ) in *Drosophila*. The authors have identified Tnc, a mucin-like protein in a genetic screen for regulators of Neto, that the Serpe lab has previously shown to be required at the NMJ. In this elegant study, the authors use a combination of physiology, immunostainings and genetic approaches to probe the pre- and post-synaptic roles of *tnc* at the NMJ. Their results show that *tnc* mutants have reduced mini frequency, EJPs and quantal content. Paired pulse facilitation experiments show that *tnc* mutants have lower release probability. Next, the authors use pre- and post-synaptic specific manipulations (knock-down and rescue) to show that *tnc* is required in neurons but not muscles for normal neurotransmitter release. They further show that presynaptic RNAi knock-down of α and β integrin genes phenocopy the *tnc* phenotypes and suggest that *tnc* may act as an integrin ligand. Next, the authors examine the effects of altering tnc levels pre- and post-synaptically on integrins and synaptic morphology at the NMJ and find that the source and dose of *tnc* are critical determinants of bouton architecture and synaptic function. S2 cell experiments nicely show the ability of Tnc to cluster integrins as also suggested by the PLA assay.

Overall, this is an interesting study that sheds key novel insights into the role of extracellular matrix ligands and receptors in shaping the NMJ architecture and function. Although in my opinion the spectrin connection does not add much to the story and the precise mechanism remains to be established (it may have to do with post-translational modifications, as suggested) this study is well executed, novel and of broad interest.

My only criticism is the absence of driver controls for phenotypic studies. While the authors use the UAS lines as background, which is fine, it is rather standard and important to control for any potential contributions of the GAL4 drivers to the phenotypes evaluated here.

---

## [Author Response]

[Editors’ note: the author responses to the first round of peer review follow.]

All three reviewers (see below their comments) expressed major criticism regarding the manuscript conclusions. The major comment is that the results are confusing and some time contradicting regarding the muscle, versus neuron-dependent function of Tenectin in recruiting integrins to the NMJ. In addition a direct versus indirect effect of Tenectin on the levels of Spectrins at the NMJ should be further validated. Many controls are also missing.We hope you find our comments useful. If you address these comments and would like to make a fresh submission at eLife, rather than elsewhere, we will be happy to examine this afresh.

I would like to express my deep appreciation to all reviewers for their careful consideration of this manuscript and for their very useful comments and suggestions. The current extensively revised manuscript includes a set of completely new ideas that were generated because of the expert comments we received from our reviewers. We have built a new, more refined model for the pre- and post-synaptic Tnc functions by carefully addressing all the suggestions from the reviewers and by expanding our study to include many additional experiments and new types of assays.

As all the reviewers pointed out, this is a complex system, distributed asymmetrically between the pre- and post-synaptic compartments, and also sensitive to the source and the dose of Tnc.

1) To clarify the contributions of pre- and postsynaptic Tnc, we first searched for additional metrics to characterize the muscle-specific functions of Tnc. On the neuron side, Tnc-dependent activities are clearly measured using very sensitive electrophysiological recordings. We found that postsynaptic Tnc is not only required for normal postsynaptic specializations, such as subsynaptic reticulum (SSR), but also for normal bouton size. Loss of muscle Tnc produces significantly smaller synaptic boutons. Bouton size appears to be a very reliable and informative read-out for Tnc activities in the muscle and was utilized throughout this revised manuscript.

2) Secondly, to deliver very low, physiologic levels of Tnc in the muscle, we first screened our large collection of individual *tnc* transgenes. Since none of the 20 lines screened had sufficiently low expression levels, we had to utilize a battery of muscle specific drivers and low temperature rearing conditions. These manipulations allowed us to expand the dose-dependent experiments and provide a comprehensive set of analyses for the Tnc function and distribution in rescue and overexpression settings.

3) Using these refined analyses, we were able to show that although Tnc associates with integrin even in tissue culture settings, only the *cis* Tnc/integrin complexes appear to have normal activity in vivo. This is a big departure from our original interpretation that extracellular Tnc can bind to integrin on either side of the synaptic cleft and engage in compartment-specific activities.

In this revised manuscript we have carefully controlled the dose of Tnc to demonstrate that:

a) Endogenous levels of muscle Tnc rescued the accumulation and function (bouton size) of the muscle Tnc/integrin complexes at *tnc* NMJs.

b) In contrast, complexes formed by neuronal Tnc and muscle integrin partially accumulate at *tnc* mutant terminals, but these *trans* complexes did not function properly as they did not rescue the bouton size and the levels of synaptic α-Spectrin.

c) In addition, we have successfully visualized a presynaptic pool of integrin – as suggested by one reviewer.

4) Genetic manipulations of *tnc, mys/βPS* and *if/αPS2* allowed us to assign these functions to the Tnc/integrin complexes and to establish that the synaptic accumulation of α-Spectrin is downstream these complexes.

5) The Tnc/integrin co-localization studies were greatly enhanced by new assays suggested by our reviewers, in particular the PLA.

6) The in vivo differences between *cis* and *trans* Tnc activities indicate that the motor neurons and the muscles secrete functionally different Tnc proteins, presumably with different posttranslational modifications. Our new Western blot analyses suggest the possibility of muscle specific processing, since a set of three faint bands are visible for Tnc overexpressed in muscle vs. mostly one band for neuronal Tnc.

7) As the reviewers predicted, refining our experimental approaches revealed an intricate regulation of the Tnc/integrin complexes. For example:

a) The neuron-derived Tnc and βPS integrin limit the synaptic accumulation of (i) muscle Tnc/integrin complexes, and (ii) α-Spectrin.

b) Excess neuronal Tnc appears to disperse the synaptic Tnc/integrin complexes (which are predominantly postsynaptic) and thus reduce the bouton size.

c) A short and weak pulse of *tnc* expression in the muscle induces elevated levels of synaptic Tnc and integrin; a longer pulse or constant muscle overexpression effectively disrupt the Tnc/integrin accumulation at synaptic terminals and produce aberrant NMJ morphology.

In summary, our revised model proposes that Tnc is a tightly regulated component of the synaptic ECM that functions in *cis* to recruit αPS2/βPS integrin and the spectrin-based membrane skeleton at synaptic terminals and together modulate the NMJ development and function. *trans* Tnc/integrin complexes can form but cannot function properly at synaptic terminals and instead appear to exhibit dominant-negative activities.

Once again, all these important insights were possible because of the thoughtful suggestions from our reviewers. We are very grateful for their consideration and help.

Reviewer #1:This paper describes the expression and function of Tenectin at the larval NMJ. While there is a lot of information in the paper, it will primarily be of interest to specialists in fly neuromuscular system development.

We respectfully disagree with the reviewer as the nature and biology of integrin ligands is poorly understood across the animal kingdom. Although Tnc has no direct human homologue, Tnc is a typical mucin, with a highly recognizable organization (two large PTS domains, multiple vWFC and RGD-like motifs). Our discovery of a synaptic mucin that binds integrin and controls synapse development and function opens the door to probe for similar mechanisms that modulate integrin recruitment and function at synapses. Furthermore, this study reveals a completely novel connection between the spectrin-based membrane skeleton and the ECM/integrin complexes. This association may inform the current research on the highly ordered spectrin network observed in mammalian neurons and dendrites.

There are also some problematical aspects to the data, as follows:1) Maybe I am not understanding something here, but there seems to be an unresolved contradiction about cis vs. trans action of Tnc. They show that muscle, but not neuronal Tnc knockdown reduces βPS expression (Figure 4M) and from this (subsection “Muscle Tnc recruits postsynaptic integrin”, second paragraph) they conclude that it muscle Tnc acts in cis to recruit βPS, which is primarily expressed in muscles.However, in Figure 6 they show that in a tnc mutant the loss of expression of βPS is restored by neuronal expression of Tnc, indicating that it is acting in trans (Figure 6D). When they express Tnc in muscles they lose βPS expression. The results in Figure 4 would predict that neuronal expression of Tnc in the mutant should not rescue, since neuronal knockdown does not affect βPS, and that muscle expression of Tnc should rescue, since muscle knockdown decreases βPS. Perhaps this is a Tnc overexpression phenotype. To test this, they should use weaker muscle drivers and reduce the temperature to reduce GAL4 activity, or make a direct muscle promoter>Tnc fusion. Unless I am missing something here, if they can express Tnc at physiological levels in the muscles in a tnc mutant, this should rescue βPS expression, not eliminate it.

The reviewer is absolutely correct and our new experiments along this line (included in Figure 7) support this view: physiological levels of Tnc in the muscle indeed rescued the defects observed in a *tnc* mutant.

We have followed the reviewer’s advice and refined our rescue experiments by using additional weak promoters, such as *BG487-Gal4*, which has graded expression (a) along the anterior posterior axis and (b) in different muscle subsets. We further reduced the Tnc expression levels by means of low rearing temperatures. Using this weak promoter and rearing the animals at 18**°**C, we managed to reduce the levels of Tnc in the muscles to ~2 fold more than endogenous levels. In these conditions we observed complete muscle rescue.

We apologize for the confusing way in which we previously described this complicated system. As recommended by reviewers, we performed and included in this manuscript a large number of experiments to better characterize the presynaptic and postsynaptic functions of Tnc, to distinguish between *cis* vs. *trans* Tnc activities, and to clearly describe the dose-dependent phenomena. As summarized above, our revised analyses demonstrate that in vivo Tnc functions in *cis*. Neuron-derived Tnc could form only unproductive complexes *in trans* with the muscle integrin.

2) The data in Figure 8 are not convincing. Looking at the high-mag views of the HA beads and placing them into the low-mag views of the cells, one can see that there is no accumulation of βPS, αPS2, or spectrin around the beads relative to their levels elsewhere in the cells. In fact the beads are in low-expression areas for all of these markers. Given this, one cannot believe that the levels of these markers are lower around the control beads. No high-mag views of the control beads are shown, and you can't even see these beads in the fluorescent images since they are not labeled. To do this experiment correctly, they need to use a different HA-tagged protein on the control beads, so that they can be seen in the fluorescent images, and show equivalent high-mag views of the beads so that it can be determined if there is really more integrin around the beads. This should also be quantitated by measuring fluorescence intensity in the bead regions in Tnc-HA vs. Control protein-HA beads.

As the reviewer suggested, we repeated these experiments using control beads coupled with an unrelated HA-tagged protein. We also included high-magnification views of the control and experiment beads as requested by the reviewer. This new set of data is presented in Figure 8 and clearly illustrates the ability of Tnc-HA coated beads, but not control beads, to recruit integrin and α-Spectrin.

3) Figure 8—figure supplement 1 is also unconvincing and incomplete. There is clearly a lot of βPS within the cell in Tnc-HA expressing cells, but levels at the cell surface are not clearly increased (at least the images shown do not demonstrate this). They could address this by staining live cells with anti-βPS without detergent. Maybe Tnc-HA is expressed at very high levels and sequesters βPS within a secretory compartment where it cannot be degraded, thus increasing its intracellular levels. This does not show that Tnc-HA acts in cis to increase integrin surface localization.

We fully agree with the reviewer regarding the limitations of this experiment. Therefore, we toned down our interpretation and included the alternative possibility that the reviewer suggested.

4) If they want to evaluate whether Tnc-HA induces βPS, they should make a stable S2 line expressing Tnc-HA, which is easily done using stable transformation vectors. If Tnc-HA expression is toxic they can drive it from a conditional promoter. Similarly, they can use stable transfection to knock down βPS with RNAi if the transient RNAi knockdown is inadequate (subsection “Tnc engages integrin and spectrin at the cell membrane”, first paragraph).

Whether Tnc-HA also induces βPS transcriptional expression is likely via an indirect mechanism. This has very little relevance for the phenomena characterized here and therefore has not been pursued.

Reviewer #2:This work by Wang and colleagues has some clear merits and high potential, but I do not recommend publication, but rather rejection with the option to re-submit. The main reason for this suggestion is the lack of clarity across the paper how experiments link together into new understanding. In my view there are seemingly contradicting results in this work that suggest a very intricate role and regulation of tnc, but have to be resolved before this manuscript can be considered for publication. The contradictions mainly consist in obscure effects observed upon pre- versus post-synaptic manipulations that require a far more refined experimental approach and logic dissection. Instead, data are being uncritically reported and the reader is left alone with a highly confusing message, if any message at all. There is potential in many of the data, but the authors failed to turn them into a readable and informative manuscript. Apart from these shortcomings there are a number of further issues that need clarification. See my detailed comments below.

We thank the reviewer very much for appreciating the clear merits and high potential of our work. As he pointed out, our findings uncovered a very intricate role and regulation of *tnc* that required a more refined experimental approach. In this expanded, revised manuscript, we have methodically addressed all the comments and suggestions of the reviewer and indeed come to a new appreciation about Tnc function and regulation. We hope this extensively revised manuscript clearly captures our current understanding and conveys our excitement about the phenomena that we have discovered and reported here.

Detailed comments:The Introduction reads well and provides a sound overview of ECM at mammalian and Drosophila synapses. It would be helpful to mention the obvious gaps of understanding in the field and how this work addresses these gaps. A tip: do the authors know this review: Singhal, N., Martin, P. T. (2011). Role of extracellular matrix proteins and their receptors in the development of the vertebrate neuromuscular junction. Dev Neurobiol 71, 982-1005 – https://doi.org/10.1002/dneu.20953

As the reviewer recommended, we modified the last paragraphs in the Introduction to summarize the gaps in the field and our contributions. The revised Introduction includes the following sentences:

“In flies as in vertebrates, the ECM proteins and their potential receptors have important roles in synapse development, but their functions have been difficult to assess at the genetic level due to the complexity of interactions of these large proteins.”

“Here, we report the functional analysis of Tenectin (Tnc), an integrin ligand secreted from both motor neurons and muscles that accumulates at synaptic terminals and functions in *cis* to differentially engage presynaptic and postsynaptic integrin.”

Introduction or start of Results: It needs to be clearly stated that tnc has no mammalian homologue.

The of reviewer is correct in that Tnc has no clear mammalian. As suggested we have stated that early in the Results (subsection “Tnc localizes at synaptic terminals”, first paragraph). However, Tnc has a highly recognizable organization typical for all mucin-type proteins.

“Using a polyclonal anti-Tnc antibody (Materials and methods) we found that Tnc signals are concentrated during late embryogenesis in the ventral cord, near the pioneer, anti-Fasciclin II (FasII) positive axons (Figure 1—figure supplement 1A, B and (Fraichard et al., 2006)): For this statement, the resolution of analyses is far too low, and I do not think that this statement is important. Delete and stay with the statement that staining it is strongly enriched in the neuropile.

We made the recommended change (subsection “Tnc localizes at synaptic terminals”, second paragraph).

“In contrast, the Tnc CNS signals were absent in a tnc mutant (tnc^EP^ – P[EPgy2]EY03355), predicted to disrupt both known tnc transcripts (Syed et al., 2012)”: Point out that Fas2 pattern is not disturbed in the mutant CNS. Obviously, there is not interaction with reported functions of integrins in this context:Stevens and Jacobs, 2002; Broadie, Baumgartner and Prokop, 2011.

We have extended our experiments to document the Fas2 pattern in control and *tnc* mutant embryo CNS (Figure 1—figure supplement 1C, D).

Subsection “Tnc localizes at the synaptic cleft”, second paragraph: indicate the two tnc transcripts in Figure 1A.

We have updated the Figure 1A to indicate the two *tnc* transcripts.

Subsection “Tnc localizes at the synaptic cleft”, second paragraph and thereafter: the correct term is "hetero-allelic": if tncDf is a true deficiency you need to talk of a "hemizygous" condition.

We have made this correction throughout the text.

Subsection “Tnc localizes at the synaptic cleft”, third paragraph”: At this early point, the pre-postsynaptic statement cannot be made. Tnc expression in the neuropile is in patches which might reflect staining around dendrites. Therefore, the expression/localisation in motorneuons may well be postsynaptic (equivalent to muscle expression in the periphery).

We removed this statement as recommended.

Subsection “Tnc localizes at the synaptic cleft”, third paragraph”: Does complete knock-down in the CNS get rid of all expression or is tnc only expressed in motorneurons but not very efficiently knocked down? Along these lines, given potentially expected different driver strengths in CNS and muscles, the conclusions about where tnc is strongest expressed cannot be drawn from the knock-down data.

We have revised these sentences and conclusions.

Subsection “Tnc localizes at the synaptic cleft”, last paragraph”: the staining does not look like being around nuclei, but seems suprisingly nuclear. If it were in Golgi or ER, it would have a broader expression.

The reviewer is right: There is a clear nuclear signal in the muscle nuclei. Upon further examination, we suspect that this signal is nonspecific, as the polyclonal Tnc antibodies label two more small bands in the larval muscle (see below). The intensities of these bands do not vary with the Tnc levels in the muscle.

We have revised the images to direct the reader’s focus on the NMJ structures and, when appropriate, on the muscle attachment sites.

Subsection “Tnc localizes at the synaptic cleft”, last paragraph”: the reduced staining is not very convincing. This needs to be quantified.

We have quantified these signals and reported the data in Figure 1—figure supplement 1E.

Subsection “Tnc localizes at the synaptic cleft”, last paragraph: either show and quantify the tnc over-expression data or take them out.

We have quantified these overexpression signals and reported the data in Figure 1—figure supplement 1E.

Figure 1—figure supplement 2 is not well described in its legend and is confusing. What nuclei are shown on the right? I guess it is muscle nuclei? But why is there no expression in those when driving tnc with 24B (bottom right image). Please, clarify, because at the moment this does not make sense. Expression data must be quantified across these experiments.

We apologize for the confusion; this supplemental figure is focused on NMJ branches and boutons of larvae expressing a *tnc-HA* transgene. We have provided a clearer description in the main text and in the revised figure legend.

Subsection “tnc mutants have impaired NMJ physiology”, first paragraph: either show larval turning assay results or take the statement out.

As suggested, we have removed these additional assay results.

Why were white flies chosen for climbing assays? At what age of flies were these experiments performed? An independent experiment with knock-down of tnc would be helpful.

These experiments were performed with 10 days-old adults; we have included this information in the figure legend. *w1118* flies have been used as controls throughout this study.

We have repeated the climbing assay for knock-down of *tnc* in motor neurons and muscle. The resulting animals had relatively normal climbing behavior, presumably due to either incomplete RNAi and/or tissue specific restricted manipulations. We did not include these new analyses as they were not informative.

Figure 2: indicate the statistical test used. Indicate in your sample number also how many events were included in the statistics: e.g. 200/13 (200 events from 13 NMJs).

We have provided the technical details for the electrophysiology experiments in the Materials and methods section.

- To calculate mEJP mean amplitudes, 100 events from each 10 or more NMJs (only one NMJ per animal was used) and were measured and averaged using the Mini Analysis program (Synaptosoft).

- Statistical analysis was performed with KaleidaGraph 4.5 (Synergy Software) using ANOVA followed by a Tukey post hoc test.

Subsection “tnc mutants have impaired NMJ physiology”, last paragraph: mention that reduced mEJC frequency is also in agreement with reduced quantal content due to less release sites.

The reviewer is correct: reduced mEJP frequency is in agreement with reduced quantal content due to fewer release sites. As recommended, we have included this alternative possibility (subsection “*tnc* mutants have impaired NMJ physiology”, last paragraph). However, based on our PPR analyses, we favor the explanation that *tnc* electrophysiological defects are primarily due to significantly decreased probability of vesicle release.

Subsection “tnc mutants have impaired NMJ physiology”, last paragraph: "as expected", provide references for this statement.

We have provided the needed reference and expanded this paragraph to build a better explanation for these data.

Subsection “tnc mutants have impaired NMJ physiology”, last paragraph: what interval length was used? This is not indicated in text or figure. Ideally, a plot with increasing length should be shown to illustrate that this is true short-term depression. In the graph indicate that you show the ratio of 1st/2nd pulse.

Paired stimuli (200 µsec, 1.9 V) were separated by duration of 50 ms.

In this revised figure legend we have included detailed information as requested by the reviewer.

Subsection “tnc mutants have impaired NMJ physiology”, last paragraph: It needs a clear explanation as to why short-term facilitation explains decreased probability. I can come up with my own speculations, but it is the task of the authors to make this clear.

As mentioned above, we have entirely revised this paragraph to better explain our results and make our conclusions clear.

“Both pre- and postsynaptic components modulate neurotransmitter release”: where does this statement come from? It is confusing without further explanation, especially since it is followed by posing this problem and since it opposes the title. Should this be the title of the section?

As recommended by the reviewer, we have removed this sentence.

Subsection “Neuronal Tnc modulates neurotransmitter release”, first paragraph: why "interestingly"? Is this not expected if presynaptic expression leads to release at NMJs and mutants show reduced frequency? Please, build a logic and solid argumentation.

We re-wrote this section as suggested.

In all graphs, indicate absence of tested significance with a symbol, such as "ns". In this way it is clear which bars were assessed that the asterisks have not simply been forgotten to insert.

We have included “ns” marks throughout this revised manuscript.

Subsection “Neuronal Tnc modulates neurotransmitter release”, last paragraph: the term "recruits" is not supported by the data; at this point you can merely state that tnc seems to act as a ligand for presynaptic integrin.

We have made the change as requested.

Figure 3G. I wonder whether only one set of experiments is sufficient here. Reproducing the data through knock-down of talin or inflated would clearly strengthen these observations.

As recommended, we extended our analyses and indeed reproduced these findings for *inflated* (αPS2) knockdown in motor neurons and muscles. The new set of data is included in the main Figure 3J-L.

Subsection “Muscle Tnc recruits postsynaptic integrin”, first paragraph: the co-localisation is not convincing and the authors must refine their argumentation. There is too much expression of integrin to make such a statement. Any correlation analyses of tnc would likely show overlap, but there is much expression of integrin that does not correlate. Even more, these data (double-labelling with HRP) clearly confirm previous observations that integrin localises also strongly to SSR, i.e. postsynpatic membrane. Most of the apparent overlap occurs therefore in areas that are irrelevant for the presynaptic function, as demonstrated by lack of effect upon postsynaptic integrin or tnc knock-down.

We have strengthened our co-localization studies by expanding these analyses, and including a proximity ligation assay, as suggested by reviewer #3. These data have been added in the revised Figure 4.

As the reviewer noted, the Tnc distribution is more restricted than that of integrin. Based on Western blot and immunohistochemistry analyses of various tagged and untagged *tnc* transgenes, we suspect that our polyclonal antibodies against Tnc, raised against an N-terminal epitope, may recognize just a portion of the synaptic pool of Tnc; the N-terminal may be removed during post-translational processing. More data describing our polyclonal antibodies and suggesting the presence of post-translation processing of Tnc in the muscle have been included in an additional supplementary figure (Figure 7—figure supplement 1).

In the course of this study we tried (1) to generate antibodies using several internal Tnc epitopes, and (2) to reproduce the successful antibodies raised against the N-terminal epitope (Frainchard et al., 2006). We only succeeded in reproducing the previously published antibodies; other internal epitopes yielded very weak if any immunoreactivities.

Figure 4: to address the presynaptic localisation of integrins, it would be very helpful to knock down integrins postsynaptically and see that staining remains in a sharp line only around boutons – which should then vanish in tnc mutant background. Please, combine G and H into one graph (both are normalised anyway)

As the reviewer pointed out, our data confirm previous reports that βPS integrin localizes strongly to SSR/ postsynaptic membrane.

We tried to follow the reviewer’s suggestion to knock-down integrin in the muscle and reveal the presynaptic pool. However, depletion of integrins in the muscle severely affected the muscle development before the loss of postsynaptic βPS was evident. Instead, we performed a similar experiment by introducing a pulse of postsynaptic Tnc and monitoring the disappearance of βPS from synaptic locations.

In brief, we reared the *BG487-Gal4/ UAS-tnc* flies at 18**°**C then moved them at 25**°**C for 4h, 8h or 12h. We found that the levels of synaptic Tnc increased at these time points but the postsynaptic βPS levels decreased, revealing a clear pool of presynaptic βPS. The 8h time point results have been included in the revised Figure 8. When these flies have been maintained at 25**°**C for longer periods, we observed an apparently complete disruption of synaptic βPS, such as reported for chronic overexpression settings (Figure 8D).

Subsection “Muscle Tnc recruits postsynaptic integrin”, second paragraph: "secreted into the synaptic cleft"?

We have changed to “secreted at synaptic terminals”.

Subsection “Muscle Tnc recruits postsynaptic integrin”, second paragraph: loss of βPS upon muscle but not neuronal knock-down of tnc is in stark contrast to data in Figure 1—figure supplement 2 where only neuronal tnc sheds into the synaptic cleft. The stabilisation theory based on ligand binding does therefore not work.

We think that the stabilization theory is still valid but it is compounded here because of the source- and dose-dependent Tnc distribution and function. When Tnc is secreted from the neurons, it first accumulates and presumably stabilizes integrin at synaptic terminals; high excess Tnc disperses away from the synaptic terminal. This may reflect limiting presynaptic levels of integrin and non-productive postsynaptic *trans* Tnc/integrin complexes, which are not stabilized and retained at synaptic terminals.

Similarly for the muscle Tnc: a low pulse of Tnc expression induces an initial increase of synaptic Tnc and integrin (as shown in the revised Figure 8G-H’); further *tnc* expression a produces a striking reduction of Tnc/integrin signals at synaptic locations. We interpret this apparent dispersion as an inability of excess Tnc to be recruited/ stabilized at synaptic terminals. Consequently, Tnc accumulates elsewhere on the muscle membrane (or in the secretory compartment) where it may sequester the integrins.

Subsection “Muscle Tnc recruits postsynaptic integrin”, second paragraph: How can the authors suddenly favour postsynaptically derived Tnc, although all their data from before show a physiological relevance only for neuronal Tnc? I am completely lost at this point.

We apologize for the confusion. We revised the sentence to specify that postsynaptic Tnc is required for the structural integrity of synaptic boutons.

As the reviewer noted before, the predominant pool of integrin at synaptic terminals is postsynaptic. We wanted to document that postsynaptic integrin depends on Tnc for its distribution. In this revised manuscript, we have assigned a clear role for the *cis* Tnc/integrin postsynaptic complex in controlling bouton size and SSR integrity. Thus, Tnc-dependent complexes assemble in both pre- and post-synaptic compartments and perform distinct functions.

Subsection “Muscle Tnc recruits postsynaptic integrin”, last paragraph: "is required for the recruitment of PAK" – the statement as is, suggests a direct mechanism.

We replace this statement to “βPS integrin precedes the recruitment of PAK” as we wish to point out the difference in the sequence of events observed at the NMJ.

Subsection “Muscle Tnc recruits postsynaptic integrin”: the last sentence is too condensed.

As suggested, the sentence was split in two.

Figure 5: I do not believe the EM data. How many muscles from how many independent animals were analysed? Were larvae processed in the same vial? Image quality is very poor and there is a clear osmolarity effect; from anti-FAK staining one gets the impression that SSR is smaller, yet here it is blown up in mutants. Analyses have to be performed in whole mounts with SSR markers (for example anti-DLG) and EM analyses have to be made more transparent to the reader and properly quantified. The odd shape of boutons looks not more than the section being almost tangential to the bouton. Shape statements are far easier made from whole mount stainings.

We set up a new collaboration towards obtaining a new set of improved electron micrographs. Consequently, this manuscript has an additional author.

We also followed all the reviewer suggestions and included whole mount stainings with Dlg to further characterize the bouton phenotypes and the SSR defects. These data have been included in the revised Figure 5.

Subsection “tnc mutants have disrupted spectrin-based membrane skeleton”, third paragraph: there are no analyses of bouton sizes in the paper, which must be provided for mutant and upon pre- and post-synaptic knockdown of tnc.

We thank all reviewers for requesting extensive analyses of bouton sizes. These detailed analyses were instrumental in characterizing the postsynaptic function of Tnc/integrin complexes and helped us clarify critical aspects of this study.

In this revised manuscript we have included multiple analyses of bouton sizes for a wide variety of genotypes.

Subsection “tnc mutants have disrupted spectrin-based membrane skeleton”: to my knowledge, loss of spectrin causes a cell-autonomous defect in neuronal terminals, i.e. the change in boutons is not caused by loss of spectrin in muscles. This does not fit with the argumentation of the authors.

The effects of loss of spectrin in both neurons and muscles have been examined in a pair of papers from Graeme Davis laboratory: Pielage et al., 2005; Pielage et al., 2006.

They report:

1) “The most striking phenotype at NMJs that lack either post- synaptic α- or β-Spectrin is the disruption of the SSR.”

“The SSR is generally thinned above and below the synaptic bouton (orthogonal to the muscle surface) and stretched laterally (parallel to the muscle surface).”

2) “Bouton number is significantly reduced in animals lacking either α- or β-Spectrin, demonstrating that postsynaptic Spectrin is necessary for normal NMJ growth.”

These phenotypes are stronger but similar to those observed in our analyses. Also, although the size of the boutons is not measured, they report reduced NMJ growth and show images with smaller boutons.

Subsection “Overexpression of Tnc disrupts postsynaptic βPS integrin and spectrin”: the manuscript gets utterly confusing now, and the authors make no effort to shed light into this. It becomes close to unreadable. There seems to be a constant contradiction between pre- and postsynaptic requirements which are not resolved.

As outlined above, we have extensively revised this section, starting with clearly delineating the pre- and post-synaptic requirements for Tnc, and providing comprehensive dose-dependent analyses. Using these refined approaches, we showed that only the *cis* Tnc/integrin complexes have normal activities in vivo. This is a big departure from our original interpretation that extracellular Tnc can bind to integrin on either side of the synaptic cleft and engage in compartment-specific activities.

The reviewer was correct all along and helped us see that our original data pointed elsewhere. We are very grateful for his tremendous help.

Subsection “Overexpression of Tnc disrupts postsynaptic βPS integrin and spectrin”, second paragraph: please, explain what the point of these experiments is? What do we learn? This is an artificial situation that does not tell us much.

The reviewer is correct: these data are less informative and do not enhance our story. We have eliminated these experiments.

Discussion, first paragraph: integrins are very well known to be locally activated through ligands, so I do not understand this claim.

We have removed this sentence, as suggested.

Reviewer #3:The manuscript "Tenectin recruits integrin to stabilize bouton architecture and regulate vesicle release at the Drosophila neuromuscular junction" represents an interesting investigation of the role of a secreted EMC protein at the NMJ. The manuscript provides novel information on the role of Tnc in recruiting the integrin complex (αPS2/βPS) to the NMJ along with spectrin and adducin. Overall the manuscript presents interesting and novel insights into the role of Tnc and integrin complexes at the NMJ. Much of the data and the quantitation was convincing and supported the conclusions. However there are a number of points that need to be addressed with the explanation of the data, some of the major conclusions drawn, a need for discussion of alternative hypotheses and the confirmation of the protein complex in vivo. These points are outlined below:

We thank the reviewer for the kind comments on the novelty and quality of our work and for the thoughtful suggestions and comments that greatly enhanced this manuscript. Because of the points that this reviewer raised we expanded our analyses and gained critical insights in the regulation of Tnc and Tnc-dependent complexes. In addition, the PLA experiments that this reviewer suggested strengthened our claim that Tnc and integrin share the same space at synaptic terminals.

For the RNAi experiments only one tnc^RNAi^ line and one mys^RNAi^ was used throughout. Neither of these lines have been previously published to be specific to Tnc or mys and thus this either needs to be proven or alternative previously verified lines used to ensure that there are not any off target effects. Specifically the experiments need at least two RNAi lines used for the Tnc and mys knockdown to ensure no off target effects or RNAi lines verified by others to be specific. The mys^RNAi^ for instance is presumably a weaker one given that muscle knockdown of mys if strong would result in embryonic lethality.

As the reviewer pointed out, we too were aware of off-targeting effects and thus tested multiple tnc-RNAi lines, including an additional line from Vienna (v42326). However, the Vienna line was very weak, as indicated by Western blots from tissue-specific knockdown animals. Similar to the Tnc-RNAi line selected for this study, *M>tnc*-v42326 induced a reduction of βPS synaptic levels (albeit much milder than *M>tnc*-TRIP), while *N>tnc*-v42326 induced robust increase of βPS levels. This behavior indicated that the phenotypes observed are due to Tnc knockdown and not to off-target effects. We have mentioned the multiple RNAi lines tested and the similar outcomes in the Materials and methods section.

We have specifically selected a weak *mys^RNAi^* line for this study, since strong knockdown of mys induces early lethality and/or profound defects in the muscle development and could be problematic when examining the NMJ development.

In addition, we have repeated the knockdown experiments and targeted if/α-PS2. In this case, we tested three available TRIP lines and found that two of them impaired the larval development when expressed in the muscle. We selected the one that allowed for apparently normal muscle development into third instar stages (BL-38958); this line reduced the synaptic αPS2 levels to ~55% of control NMJs, and α-Spectrin to ~65%. The results for if/αPS2 knock down have been reported in Figures 3J-L.

In Figure 3 comparing mEJP Frequency for the tncEP/Df was less strongly affected compared to the N>tncRNAi in Figure 3E versus 3B. It is surprising that the RNAi would have a strong effect that the EP/Df combination especially given that in Figure 1 the protein levels were substantively reduced in the EP/Df but not the N>tncRNAi. This disconnect between levels of protein versus phenotypes need to be addressed in the text.

(Figure 1) is likely due to Tnc expression in other brain cells. In this revised manuscript, we have refrained from discussing unrelated Tnc expression patterns because of the complexity of the phenomena already described here.

Since all our data indicate that the neuronal knock down of Tnc is very efficient and induces strong postsynaptic responses (see below), we alluded to this early on and mentioned additional Tnc-expressing cells in the brains with the following sentence:

“Neuron specific RNAi knockdown reduced the Tnc levels in larval brains to 43% of the control group; this generated very strong phenotypes (below) suggesting that the residual band could reflect additional Tnc-expressing cells in the larval brain.”

With the residual Tnc immunolabeling present in the tncEP/Df, it is logical to conclude that there is significantly more Tnc present in the N>tncRNAi NMJ. However the degree of reduction of Tnc at the NMJ is not shown with either N>tncRNAi or M>tncRNAi. This should be included or if not, explained. This becomes very relevant in Figure 4 as the effect of M>tncRNAi on the recruitment of αPS2βPS to the NMJ had a stronger effect than N>tncRNAi.

This request (together with a similar one from reviewer #2) prompted us to carefully quantify the synaptic Tnc in a number of settings, including the RNAi knockdown experiments. The RNAi results are now reported in a completely new figure (Figure 4—figure supplement 2).

Because of this careful quantification we uncovered a completely new layer of Tnc regulation, a push-pull mechanism, where removal of neuronal Tnc (in an otherwise wild-type background) induces a significant increase of the muscle-derived Tnc. This increased muscle Tnc triggers a downstream cascade that includes (a) increased integrin (Figure 4N), (b) increased Dlg (Figure 5D) and (c) increased α-Spectrin (Figure 6—figure supplement 1). We were also able to show that similar manipulations of βPS have exactly the same effect as Tnc on the synaptic α-Spectrin, that is, neuronal knockdown of βPS triggered almost doubled α-Spectrin levels (Figure 6—figure supplement 1).

The degree of Tnc loss in the tncEP/Df combination in Figure 1G (and Figure 1—figure supplement 1D) was surprisingly not complete given that the allele seems to be a null. The immunolabeling in the image does not match the quantitation by Western seen in Figure 1C. The phrase "much reduced" in tncEP/Df could be quantified using the approaches utilized throughout the manuscript for all the other markers. The text should also address if this is truly the presence of residual Tnc at the NMJ (and the implications of this) or is this background?

We have addressed this issue by including a careful quantification of the NMJ signals (Figure 1—figure supplement 1) as described above. Furthermore, we have provided a completely new figure (Figure 7—figure supplement 1) showing a Western blot of brain and muscle extracts of various genotypes labeled for Tnc. In the brain protein extracts, the Tnc antibodies label a single band of the expected size (300 kDa); in the muscle extracts, we detect two more bands that are independent of various *tnc* manipulations. This new data indicate that our Tnc antibodies show some non-specific staining in the muscle.

The results from this manuscript suggest that αPS2βPS and Tnc may directly interact. The S2 cell experiment was a strong approach to understanding the associations of Tnc, the integrin complex, spectrin and adducin. A stronger and a necessary approach to investigating the association of these proteins was to prove that these occur in vivo using a proximity ligation assay (PLA) to determine where in the NMJ these associations happen. The advantage of this approach is that the tncEP/Df mutant would serve as a good control as would the different RNAi and overexpression approaches. The PLA experiment would also be able to address if the M>tnc expression was able to recruit the αPS2βPS complex to ectopic sites away from the NMJ. This would go a long way to support many of the conclusions and how a clear link at the NMJ between these protein complexes.Further to this point, the link between integrins to spectrin and adducin is the key point of interest but other than the S2 cells there is little in the way to support a physical link between these components. It is equally possible that the loss of spectrin or adducin itself may lead to the loss of αPS2βPS from the NMJ. What happens to spectrin or adducin when αPS2 is reduced and vice versa?

We thank the reviewer very much for this suggestion. Indeed, we have successfully exploited the PLA assay to document the close proximity of Tnc and βPS at synaptic terminals. These new data are included in Figure 4G-H’.

As mentioned above, we have performed and included some epistasis experiments in this revised manuscript. For example, neuronal knockdown of Tnc triggered a significant increase in postsynaptic Tnc, β-PS and α-Spectrin (Figure 4, Figure 4—figure supplement 2, Figure 6—figure supplement 1). Similarly, neuronal knockdown of βPS triggered almost doubling of the α-Spectrin synaptic levels (Figure 6—figure supplement 1). Muscle knockdowns of βPS or αPS2 were less informative for weak RNAi lines (Figure 6—figure supplement 1 and not shown), and induced overall defective muscle morphology and compound phenotypes for strong RNAi settings.

For Figure 5 it is mentioned that "In the course of these experiments we noted that tnc mutant NMJs showed aberrant morphology with poorly defined bouton/interbouton boundaries and more tubular branches, particularly in the proximal region." Given that this is the first mention of these different phenotypes, an inclusion of some examples is warranted as prior indications from Figure 2—figure supplement 1 was there were little morphological changes.For the TEM analysis comments such as "However, the mutant boutons were drastically distorted and no longer maintained normal round/oval shapes" and "This phenotype was highly penetrant and affected all type 1B neurons" should be supported by the number of boutons that were analyzed and the number of synapses/larvae analyzed in the text itself.

As emphasized above, the request to document in details the NMJ morphology turned into a tremendously informative readout for the Tnc function in the muscle. In particular, the bouton size and the SSR density appear to be a direct consequence of the activity of postsynaptic Tnc/integrin complexes. Loss of muscle Tnc invariably led to smaller boutons with reduced SSR structures and Dlg signals; these defects could only be rescued by the *cis* Tnc/integrin postsynaptic complexes.

As indicated in the response to reviewer #2, we have now included a new set of EM data with extensive quantifications.

The distribution of Dlg should be shown (data not shown throughout the manuscript) and the quantitation provided, especially given the effects on adducin on Dlg at the NMJ. This later point should be addressed in the text as well.

We thank the reviewer for this suggestion. as careful quantifications of the Dlg levels greatly contributed to the readout for the Tnc function in the muscle.

The M>tncRNAi data with α-spectrin showed no decrease compared to the UAS-tncRNAi control. While the N>tncRNAi showed an upregulation. Yet the tncEP/Df displayed a reduction in α-spectrin. The manuscript needs to address these differences and provide a model to help guide through the different interpretations of these effects.

As outlined above, we have extensively revised this manuscript using compartment specific and dose-dependent analyses. Using these refined approaches, we showed that only the *cis* Tnc/integrin complexes are biologically functional in vivo, whereas the *trans* Tnc/integrin complexes exhibit dominant-negative activities. This is a significant departure from our previous interpretation that extracellular Tnc can bind to integrin on either side of the synaptic cleft and engage in compartment-specific activities.

In this model, the differences between α-spectrin levels in tnc mutants and *N>tnc^RNAi^* reflect an inhibitory role for neuronal Tnc onto the muscle Tnc/integrin complexes (and consequently spectrin recruitment).

The reviewers were correct all along and rightly confused by our original interpretation. We are very grateful for their tremendous help in sorting out this complex system.

The manuscript makes the statement "Since neuron-derived Tnc recruits βPS integrin in the motor neurons to modulate neurotransmitter release (Figure 3)" – this is a very strong statement given that N>tnc^RNAi^ showed a mild increase of the distribution of βPS to the NMJ (Figure 4L, 4N) and the effect on αPS2 is not shown. Figure 3 provides no evidence that the changes observed with N>tnc^RNAi^ are due to the loss of βPS. This would require a rescue experiment to provide a direct link.

This claim is now further supported by several new pieces of data:

1) Figure 3 now includes a similar effect (reduced mini frequency) when if/αPS2 was knocked down in the neurons but not in the muscle. Thus both integrin subunits, αPS2 and βPS, are required in the neuron for normal mini frequency.

2) Genetic manipulations suggested by reviewer #2 allowed us to visualize a presynaptic pool of βPS (Figure 8H).

The reviewer is correct: the ideal experiment would require some rescue. But since only the *cis* Tnc/integrin complexes appear to be biologically active, overexpression of neuronal βPS and/or αPS2 are not expected to rescue the mini defects of *tnc* mutants.

The interpretations and conclusions for the role of neuronal or muscle driven Tnc need to be tempered. N>tnc rescued the degree of βPS present at the NMJ in the tncEP/Df mutant while M>tnc did not and this lead to the conclusion that neuronal Tnc is key to the recruitment of βPS. Yet N>tnc alone decreases the βPS at the NMJ (Figure 7C, 7E) compared to controls while N>tncRNAi knock down (Figure 4L, 4N) shows slight more βPS at the NMJ compared to control. This suggests that neuronal derived Tnc blocks integrin accumulation at the NMJ. On the muscle side of the equation M>tnc cannot rescue the tncEP/Df which isn't too surprising given that M>tnc removes all the tnc from the NMJ. Thus these experiment don't prove that neuronal derived Tnc is sufficient and necessary to recruit βPS. Especially given that Tnc knockdown in muscles but not in neurons reduced the amount of synaptic βPS.

The reviewer is absolutely correct and examining these points in more detail made us depart from our original thinking that extracellular Tnc can bind to integrin on either side of the synaptic cleft and engage in compartment-specific activities. Moreover, as the reviewer noted, neuronal derived Tnc limits the integrin accumulation at the NMJ. We have expanded our data set to further characterize this regulation and, as detailed above, found that removal of neuronal Tnc induces a significant increase of the muscle-derived Tnc. The elevated Tnc muscle levels next induce an increase in the synaptic accumulation of (a) integrin (Figure 4N), (b) Dlg (Figure 5D) and (c) α-Spectrin (Figure 6—figure supplement 1).

Along these lines the tnc mutants with muscle overexpressed tnc had a dramatic reduction in βPS as well as the M>tnc in a wild type background. The panels from Figure 1J, Figure 1—figure supplement 1F and Figure 1—figure supplement 2F would suggest that overexpression of tnc may be significantly deleterious to the entire NMJ not just the distribution of βPS.

The reviewer correctly noted the toxic effect of excess muscle Tnc. We have refined these analyses by examining a wide range of Tnc doses in muscles rescue (Figure 7) and overexpression (Figure 8) experiments.

The authors state that expression of N>tnc in the tncEP/Df mutant was able to rescue the reduction in α-spectrin however the degree of increase in α-spectrin does not appear to be strong. For the quantitation (Figure 6J) it was not clear what was being compared – each experimental to control which would suggest that the N>tnc was significantly different from control (and didn't rescue to a great extent) – or each experimental to each other which would suggest that all three experimental were significant different from each other which didn't appear to be the case. For this type of multiple experiment analysis it would also be more relevant to carry out a One Way Anova with a post hoc multi-comparison test rather than a students' t-test.It was also intriguing that the degree of rescue of Adducin and βPS immunolabeling at the NMJ was so much better than that for α-spectrin with the N>tnc. Given the strength of the statements made in the text is it possible the wrong data was included in Figure 6J?The major conclusion from this section should be tempered to reflect that neuronal expression of Tnc can rescue the tncEP/Df spectrin, βPS and adducin levels. Whether M>Tnc is able or not will require a more measured approach to ensure that muscle expressed Tnc is present at the NMJ, that the muscles themselves are not deleteriously affected. This might be case given the stronger effects on βPS by the increased expression of Tnc in wild type muscles compared to the tncEP/Df muscles. These effects may simply by an increase the deleterious effects of Tnc expression and thus lead to a great disruption of NMJ morphology rather than an specific effect on "ligand redundancy and/or other mechanisms may allow for partial βPS and/or α-Spectrin synaptic accumulation". Would the increased expression of Tnc or any ECM component lead to disruption of muscles as a result from ER stress?

As the reviewer noted, neuronal Tnc did not really rescue of the α-spectrin levels at tnc mutant NMJs. In the revised manuscript, we further exploit this observation together with the size of type Ib boutons to argue that *trans* Tnc/integrin complexes could form but are not biologically active.

We have revised extensively the rescue experiments and provided expanded data analyses, including graded expression levels for muscle Tnc. As the reviewer indicated, we have now convincing evidence that excess Tnc levels in the muscle induces deleterious effects; we have revised the text accordingly.

The statement "Unlike βPS, which was completely lost at these NMJs" is not supported by the images presented in Figure 4E, Figure 6B and 6E: βPS is not completely lost at these NMJs. The authors need to be more careful about making this type of absolute statement.

We have revised this incorrect statement.

[Editors' note: the author responses to the re-review follow.]

Reviewer #1:Wang et al. report an interesting finding on distinct pre-synaptic and post-synaptic roles for tenectin (Tnc), a selective integrin ligand, at the Drosophila neuromuscular junction (NMJ). Tnc in the neuron interacts with βPS/αPS2 integrin to control neurotransmitter release, whereas Tnc in the muscle interacts with post-synaptic αPS2/βPS integrin to regulate bouton morphology. By manipulating Tnc, they uncovered a novel role for integrin in recruiting the spectrin based membrane skeleton at the NMJ.This paper will be of broad interest to the readership of eLife because it advances our understanding of NMJ development and synaptic physiology, as well as integrin/extracellular matrix biology. Although Tnc has been studied at the Drosophila NMJ, Wang et al. is among the first to reveal that Tnc has differential function in neurons and muscle. The data presented in this paper is thorough, with good controls and overall high-quality figures. However, I have two major concerns that should be addressed before publication. First, the inclusion of only one RNAi line targeting Tnc is insufficient, especially given the fact that the neuronal RNAi line only knocks down Tnc levels to 43% of control. The inclusion of at least two RNAi lines showing the same phenotype would both confirm and strengthen the author's conclusions. This is not a detail and needs to be addressed.Second, the conclusion that αPS2, βPS, and Tnc function together to modulate neurotransmitter release (Figure 2) should be strengthened by genetic interaction experiments between βPS and Tnc indicating that either a) double mutants do not have a more severe phenotype, or b) trans-heterozygotes show enhancement of phenotype.

We are deeply grateful to this reviewer for the thoughtful consideration of our work and for her/his generous help in improving the overall presentation of our findings, and their biological relevance. Although Tnc has been studied in the context of epithelial tube morphogenesis and development of male genitalia, this is the first study reporting Tnc distribution and function at the *Drosophila* NMJ.

In the revised manuscript, we have addressed both reviewer’s concerns, as follows:

1) As the reviewer emphasized, because of frequent off targeted effects, one must always test two different RNAi lines. We have previously tested a second RNAi line and observed similar, though relatively milder phenotypes. Here we have expanded these knockdown experiments and included a complete set of electrophysiological recordings in a new supplementary figure (Figure 3—figure supplement 1). Together these data confirmed that we are observing Tnc-specific phenotypes.

2) The reviewer is absolutely correct, we should see a genetic interaction between tnc and αPS2 or βPS. Indeed, we tested the trans-heterozygotes (*mys*/+;; *tnc*/+) and observed severe defects that resembled the *tnc* mutant phenotypes, even though when tested separately both heterozygotes have no detectable defects. We have included these additional experiments in the main Figure 3M-O.

Reviewer #2:Tenectin recruits integrin to stabilize bouton architecture and regulate vesicle release at the Drosophila NMJ' Wang et al.In this manuscript the authors identify the mucin class protein tenectin as an ECM component of the Drosophila NMJ synapse from an interaction screen with Neto. The description of tenectin mutants is comprehensive, the technical quality of the data is high and the authors do identify interesting interactions with integrins. However ultimately little new insight is gained into the role of the ECM in synapse development or function beyond the identification of a new component. As such, the manuscript seems better suited to a more specialised neuroscience journal.

We respectfully disagree with reviewer #2: Tnc is a strikingly selective integrin ligand that allowed for unprecedented insights into our understanding of integrin recruitment and function at synaptic locations. Our genetic manipulations produce sharp phenotypes that uncovered novel pre- and post-synaptic functions for integrin. Also, a new signaling pathway that couples ECM/integrin with the spectrin-based membrane skeleton emerges from this work. Integrin and spectrin have pleiotropic roles in both pre- and post-synaptic compartments, and local, synaptic disruption of integrin or spectrin have not been previously possible. Our discovery provides the means to explore their local functions and to define a synaptic role for the spectrin-based membrane skeleton, a topic of intense research in neural development.

Reviewer #3:The manuscript by Wang et al. reports a novel function of tenectin (tnc) at the neuromuscular junction (NMJ) in Drosophila. The authors have identified Tnc, a mucin-like protein in a genetic screen for regulators of Neto, that the Serpe lab has previously shown to be required at the NMJ. In this elegant study, the authors use a combination of physiology, immunostainings and genetic approaches to probe the pre- and post-synaptic roles of tnc at the NMJ. Their results show that tnc mutants have reduced mini frequency, EJPs and quantal content. Paired pulse facilitation experiments show that tnc mutants have lower release probability. Next, the authors use pre- and post-synaptic specific manipulations (knock-down and rescue) to show that tnc is required in neurons but not muscles for normal neurotransmitter release. They further show that presynaptic RNAi knock-down of α and β integrin genes phenocopy the tnc phenotypes and suggest that tnc may act as an integrin ligand. Next, the authors examine the effects of altering tnc levels pre- and post-synaptically on integrins and synaptic morphology at the NMJ and find that the source and dose of tnc are critical determinants of bouton architecture and synaptic function. S2 cell experiments nicely show the ability of Tnc to cluster integrins as also suggested by the PLA assay.Overall, this is an interesting study that sheds key novel insights into the role of extracellular matrix ligands and receptors in shaping the NMJ architecture and function. Although in my opinion the spectrin connection does not add much to the story and the precise mechanism remains to be established (it may have to do with post-translational modifications, as suggested) this study is well executed, novel and of broad interest.My only criticism is the absence of driver controls for phenotypic studies. While the authors use the UAS lines as background, which is fine, it is rather standard and important to control for any potential contributions of the GAL4 drivers to the phenotypes evaluated here.

We thank very much the reviewer for the generous comments and support for our work.

As the reviewer requested, we have revised the manuscript to include the characterization of the drivers. These additional data are included in Figure 3—figure supplement 1.